# Training a Scientific Reasoning Model for Chemistry

**Siddharth M. Narayanan**[1]**, James D. Braza**[1]**, Ryan-Rhys Griffiths**[1]**, Albert Bou**[1]**,**
**Geemi P. Wellawatte**[1]**, Mayk Caldas Ramos**[1]**, Ludovico Mitchener**[1]**, Michael Martin Pieler**[1]**,**
**Samuel G. Rodriques**[1][*]**, Andrew D. White**[1][*]

[1]FutureHouse Inc., San Francisco, CA
[*]These authors jointly supervise technical work at FutureHouse.
Correspondence: {sam,andrew}@futurehouse.org

## Abstract

Reasoning models are large language models that emit a long chain-of-thought before answering, providing both higher accuracy and explicit reasoning for their response. A major question has been whether language model reasoning generalizes beyond mathematics, programming, and logic, where most previous work has focused. We demonstrate that reasoning models can be post-trained for chemistry without additional domain pretraining, and require substantially less data compared to contemporary domain-specific models. We report `ether0`, a 24B parameter LLM (based on `Mistral-Small-24B`) that can reason in natural language and respond with chemical structures. This reasoning model was trained with reinforcement learning on 640,730 experimentally-grounded chemistry problems across 375 tasks ranging from synthesizability, to blood-brain barrier permeability, to human receptor activity, to scent. Our model exceeds general-purpose chemistry models, frontier models, and human experts on molecular design tasks. It is also more data efficient relative to specialized models. We anticipate that this method can be applied to train data-efficient language models specialized for tasks across a wide variety of scientific domains.

## 1 Introduction

The dominant approach to improve the accuracy of large language models (LLMs) in recent years has been to scale pre-training corpora size and pre-training compute budget [1, 2, 3, 4]. Partly driven by the finite availability of pre-training data, however, attention has shifted towards alternative scaling dimensions. Such dimensions include strategies such as majority voting [5, 6], "budget-forcing" [7], and test-time training [8], which attempt to scale inference compute. Broadly, reasoning models attempt to improve performance emitting their thought process before arriving at an answer. Early approaches in this vein attempted to elicit reasoning behavior through chain-of-thought (CoT) prompting [9, 10]. More recently, however, reasoning behavior has been demonstrated to emerge through reinforcement learning (RL) post-training, without the need for CoT-style prompting.

RL post-training represents a shift of focus from pre-training data to problems with verifiable rewards. Solutions to such problems can be checked for correctness, allowing the model to generate new, verifiable outputs during learning, explore the space of solutions, and overcome limits imposed by fixed data resources. Multiple works have demonstrated the potential of this approach, particularly in the domains of mathematics and programming. These include both closed-source models [11, 12], and more recently, a large number of open-source models [13, 14, 15, 16, 17, 18, 19].

39th Conference on Neural Information Processing Systems (NeurIPS 2025).

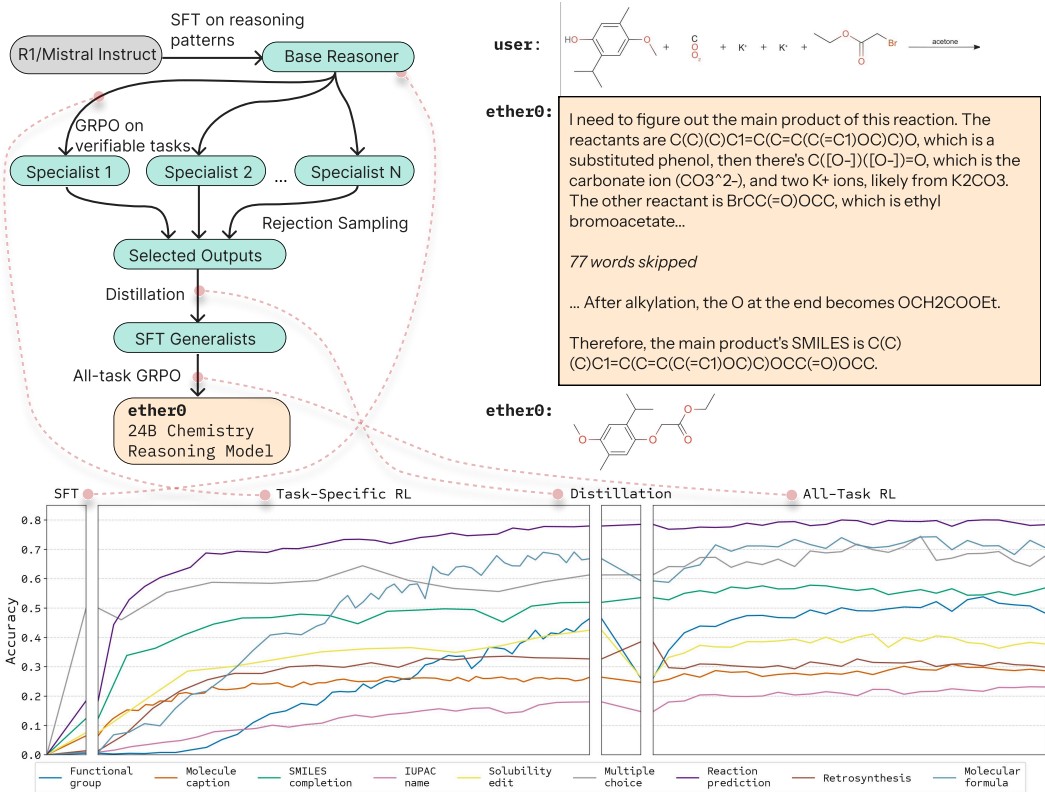

Figure 1: An overview of the training methodology and an example reasoning trace for `ether0`. Training stages are shown in the bottom panel where the accuracy per step is scaled to have the same x-axis range (see Appendix E).

Scientific domains may be particularly well suited for reasoning models because, as in mathematics and programming, it is often straightforward to assess the quality of a solution, but much more difficult to generate a solution. For example, we may be able to measure the solubility of a given molecule, yet designing a molecule with a desired solubility can be a significant challenge. These "inverse problems" are common in many areas of the physical sciences [20, 21, 22, 23, 24]. More broadly, the scientific method is grounded in structured reasoning: formulating a hypothesis based on observation, evaluating the logical implications of the hypothesis based on experiment, and refining the hypothesis based on analysis of the results of experiment. Science often involves cognitive strategies such as breaking problems into subproblems, responding to failures, or reasoning backwards from desired outcomes, which are strategies also exhibited by reasoning models [25]. However, despite the conceptual alignment between science and reasoning models, there is still relatively little work on scientific reasoning models, aside from benchmarks on multiple choice questions [26, 27, 28].

In this work, we focus on chemistry, with tasks centered on designing, completing, modifying, or synthesizing molecules. This setting is a good demonstration for *scientific* reasoning models. First, molecules can be represented in text in the SMILES format [29, 30, 31], avoiding the complexities of training a modality-specific encoder. Second, text-based representations of molecules are short relative to modalities in materials science and biology such as nucleotide sequences or CIF files. Third, generating and editing molecules is a critical application, where novel compounds may lead to meaningful clinical and commercial advancements.

We demonstrate the efficacy of reasoning models in chemical tasks by introducing `ether0`, a novel model that reasons in natural language and outputs molecular structures as SMILES strings. On the chemical reasoning tasks under consideration, `ether0` outperforms frontier LLMs, human experts, and models trained for general chemistry. Moreover, `ether0` supports key stages of the drug discovery pipeline: it can generate candidates during hit discovery, it enables molecule editing in hit-to-lead to improve potency, selectivity, or physicochemical properties, and it contributes to lead optimization by

refining compounds to enhance efficacy, reduce toxicity, and improve ADMET profiles, all while being aware of synthesizability.

To efficiently train our model, we utilize a series of optimizations over vanilla RL, including distillation of reasoning behaviors, a dynamic curriculum, and initializing RL with distillation from expert models. We further analyze `ether0`'s data efficiency, failure modes, and reasoning behavior to understand the utility of a reasoning in solving chemistry problems.

### Related Work

**Reasoning Models**   Reasoning models are characterized by an attempt to impart system 2-type decision-making [32] to LLMs. Early efforts to this affect include chain-of-thought (CoT) [9], zero-shot CoT [10], and Tree of Thought (ToT) [33] which seek to elicit reasoning by modifying LLM prompts. Later attempts make use of process-level supervision to provide feedback on individual reasoning steps [34, 35, 36]. Most recently, a number of reasoning models have been released [12, 37, 13, 11, 17, 38, 39] using large-scale reinforcement learning via Group Relative Policy Optimization (GRPO) [40] or inference time scaling [41, 7].

**Reasoning Models in Chemistry**   While frontier reasoning models have been evaluated on chemistry tasks [18, 37, 11], the vast majority of these benchmarks have consisted of chemical "knowledge" tasks rather than chemical reasoning tasks [13]. While datasets such as GPQA-D [28], MMLU [26], MMLU-Pro [27], OlympiadArena [42], and Humanity's last exam [43] assess chemistry knowledge, they do not assess the model's ability to perform sophisticated chemical reasoning tasks such as retrosynthesis and proposing new structures. While many works have evaluated non-reasoning LLMs on chemical reasoning tasks [44, 45, 46], used LLMs as components for chemical tasks [47, 48, 49], or investigated CoT-style prompting strategies [50, 51], to the best of our knowledge there have been no attempts to directly train reasoning models to perform chemical reasoning tasks using large-scale reinforcement learning. In terms of other scientific domains, *OmniScience* [52] targets general science applications through distillation on reasoning traces. *Med-R1* [53] applies GRPO to medical vision-language tasks, using reinforcement learning to improve generalization and clinically grounded behavior across multi-modal diagnostic reasoning tasks. *BioReason* [54] integrates a DNA foundation model with an LLM and combines supervised fine-tuning and GRPO to enable interpretable, multi-step genomic reasoning.

## 2   Chemical Reasoning Tasks

We construct a dataset of 640,730 chemical reasoning problems, comprising 18 different tasks. Molecules are represented in the question and expected answer as SMILES, which encodes the molecular graph or chemical reaction as ASCII text [55]. The answers are all either a molecule or a reaction. Many tasks are broken down into subtasks. For example, in the solubility editing task, one subtask is to increase solubility without changing the molecular scaffold, and another is to change it without affecting specific functional groups. Table 1 summarizes all problems in our dataset, and Section C.2 provides full details on the dataset provenance as well as the construction of each task.

We strove to only use synthesized molecules when constructing our dataset, in contrast to previous work in cheminformatics based on "hypothetical" molecules [82]. Thus, all the questions and answers are based on the result of physical experiments. Full reward function implementation details are provided in Section C.3. In addition to the criteria listed, tasks marked with $\dagger$ also check that the proposed molecules are plausibly synthesizable by fragmentation into rings and local groups (details in Section C.3).

**Solubility edit**$^\dagger$**:** Modify a given molecule to increase or decrease aqueous solubility ($\log S$). Subtasks impose additional constraints enforcing similarity to the input molecule. The $\log S$ objective is computed using KDESol [56] and constraints are evaluated using RDKit [83] and exmol [84].

**IUPAC name:** Given an IUPAC name of a molecule, produce the corresponding SMILES string for the molecular structure. Verified with RDKit.

**SMILES completion**$^\dagger$**:** Given a SMILES string of a molecular fragment, provide a completion that results in a valid molecule. Verified with RDKit.

**Molecular formula**$^\dagger$**:** Propose a molecule given a molecular formula in Hill notation [85]. Verified

Table 1: Breakdown of verifiable reward training tasks. ML model verifier: trained predictive model; MCQ: multiple-choice questions. Templates: unique phrasings per category. Data source: short name (see citations for full attribution). *Not a sum; multiple-choice property questions share templates. †Also performs a "reasonable molecule" check.

| Task | Subtasks | Examples | Verifier | Templates* | Data source name |
|---|---|---|---|---|---|
| Solubility edit | 3 | 115977 | ML model[56], code† | 15 | ChEMBL[57] |
| IUPAC name | 1 | 74994 | code | 10 | COCONUT[58, 59] |
| SMILES completion | 1 | 74990 | code† | 10 | COCONUT[58, 59] |
| Molecular formula | 1 | 18738 | code† | 10 | COCONUT[58, 59] |
| Functional group | 1 | 74562 | code† | 6 | ChEMBL[57] |
| Elucidation | 1 | 74164 | code† | 10 | COCONUT[58, 59] |
| Retrosynthesis | 1 | 67252 | ML model[60], Bloom filter[61] | 8 | - |
| Reaction prediction | 1 | 61205 | code | 10 | ORD[62, 63] |
| Molecule caption | 1 | 54148 | code | 8 | LlaSMol[64] |
| Safety | 11 | 5687 | MCQ | 8 | Pubchem[65] |
| Scent | 180 | 4240 | MCQ | 8 | pyFUME[66, 67, 68, 69, 70, 71, 72, 73, 74, 75] |
| Blood-brain barrier | 2 | 2064 | MCQ | 8 | BBB[76] |
| Receptor binding | 150 | 1663 | MCQ | 8 | EveBio[77] |
| ADME | 12 | 1030 | MCQ | 8 | Fang ADME[78] |
| Aqueous solubility | 2 | 464 | MCQ | 8 | AqSolDB[79] |
| LD50 | 2 | 342 | MCQ | 8 | Pubchem [65] |
| pKa | 4 | 336 | MCQ | 8 | IUPAC[80] |
| Photoswitches | 1 | 23 | MCQ | 8 | Photoswitches[81] |
| **Total** | 375 | 640,730 | 9 | 81 | 13 |

with RDKit.

**Functional group†:** Propose a molecule given a molecular formula and 1-3 desired functional groups. Verified with RDKit and exmol.

**Elucidation:** Determine the chemical structure of a molecule found in an organism given its molecular formula and background information on the organism. Since the problem is underdetermined, we consider any answer to be correct if the proposed molecule has a Tanimoto similarity (ECFP4 [86]) of at least 0.7 to the ground truth. Verified with RDKit.

**Retrosynthesis:** Provide a single-step reaction to produce the given target molecule. The reactants must all be purchasable molecules (determined by manufacturer catalogs in a Bloom filter [61]), and the product of the proposed reaction must match the target molecule predicted using the Molecular Transformer model [60].

**Reaction prediction:** Given a chemical reaction, predict the major product. Verify exact molecule match with RDKit.

**Molecular caption:** Given a textual description of a molecule, produce the SMILES of the molecule. This task uses data from Yu et. al [64], which itself comes from PubChem [87, 88, 89]. Verified with RDKit.

**Multiple choice questions:** Predict or modify properties of a molecule, for which no accurate oracle exists. Instead, multiple options are presented, and the model is expected to select the one that has been experimentally determined to satisfy the criterion. Verified by string matching. See Section C.1.

## 3 Background

**Supervised Fine-Tuning.** As in prior work [90, 13], we use SFT to initialize a policy for RL (Equation S1). If the demonstration dataset $\mathcal{D}_{\text{demo}}$ is itself from another policy $\pi'$, this can also be considered a form of expert iteration [91, 92] or knowledge distillation [93].

**Reinforcement Learning.** While SFT can be used to warm-start the policy, we rely heavily on online reinforcement learning to improve our models. In particular, we use Group Relative Policy Optimization (GRPO) [40].

Given a question $x$ from the dataset, we sample $G$ completions $y_1, \ldots, y_G \sim \pi(\cdot|x)$. Each is assigned a reward $r_1, \ldots, r_G$ and a corresponding advantage:

$$A_i = \frac{r_i - \text{mean}\{r_1, \ldots, r_G\}}{\text{std}\{r_1, \ldots, r_G\}}. \tag{1}$$

Given a single problem $x$ and a group of completions $\{y_i\}$, the per-group objective is:

$$J(\theta, x, y_1, \ldots y_G) = \sum_{i=1}^{G} \frac{1}{|y_i|} \sum_{t=1}^{|y_i|} \left\{ \text{clip}\left( \frac{\pi_\theta(y_{i,t}|x, y_{i,<t})}{\pi_{\theta_{\text{old}}}(y_{i,t}|x, y_{i,<t})}, A_i, \epsilon \right) - \beta \hat{D}_{\text{KL}}[\pi_\theta||\pi_{\text{ref}}; x, y_{i,\leq t}] \right\}, \tag{2}$$

where $\pi_\theta$ is the policy being optimized, $\pi_{\theta_{\text{old}}}$ is the policy from which we sampled rollouts, and $\pi_{\text{ref}}$ is a reference policy. $\text{clip}$ is the standard PPO clip function [94]:

$$\text{clip}(r, A, \epsilon) = \min\{r \cdot A, \max\{\min\{r, 1+\epsilon\}, 1-\epsilon\} \cdot A\}. \tag{3}$$

The global policy objective we seek to optimize over the training set of problems $\mathcal{D}$ is therefore:

$$\mathcal{J}_{\text{GRPO}}(\theta, \mathcal{D}) = \frac{1}{|\mathcal{D}|} \sum_{x \in \mathcal{D}} J(\theta, x, y_1, \ldots, y_G)\Big|_{y_1, \ldots, y_G \sim \pi_{\theta_{\text{old}}}(\cdot|x)}. \tag{4}$$

For completeness, the GRPO algorithm is detailed in Algorithm 1.

## 4 Training

In this section, we describe a method to train a large language model to reason about and answer the problems detailed in Section 2. We utilize a multi-stage training procedure, consisting of alternating phases of (a) distillation [93] and (b) GRPO [40, 13]. At a high level, the stages are: (1) Supervised fine-tuning on long chain-of-thought reasoning sequences; (2) Task-specific "specialist" GRPO; (3) Distillation of specialist models into an all-task "generalist" model; and (4) Generalist GRPO. Using a family of task-specific reasoning models to generate synthetic data for a generalist model has been recently demonstrated to be an effective strategy in other domains [95, 96].

Unless otherwise stated, our policies are trained from `Mistral-Small-24B-Instruct-2501` [97]. To simplify formatting of the model output, we introduce four new tokens to the base model's vocabulary to demarcate reasoning and answering boundaries. During distillation and RL, these tokens are used to respectively format and validate sequences with the following structure:

```
<|think_start|>THOUGHT<|think_end|>
<|answer_start|>ANSWER<|answer_end|>
```

### 4.1 Long CoT Supervised Fine-Tuning

We warm-start our model with SFT on rejection-sampled long chain-of-thought sequences to jump-start RL with a policy for which reasoning and SMILES answers are already in-distribution.

The SFT sequences are first generated by prompting DeepSeek-R1 with a subset of the training dataset, with a maximum token budget of 8192 tokens. To remove low-quality responses, we enforce the following criteria: (1) each sequence ends with an answer enclosed in XML tags; (2) the answer is valid SMILES/SMIRKS; and (3) passes an LLM-based check for relevant reasoning (Section B.1). We considered rejecting responses with incorrect answers, but R1's success rate is below 1% for many tasks. Our goal during SFT is to find a good pre-RL initialization, not necessarily to maximize accuracy, and prior work [13, 25] suggests that SFT even on inaccurate reasoning sequences can be sufficient. Therefore, we do not discard sequences that end in incorrect answers.

Early experiments showed that starting RL with long reasoning sequences was inefficient: sampling dominates training time, and the extra reasoning did not translate to higher accuracy. So instead, we prompt `Mistral-Small-24B-Instruct-2501` to summarize R1-generated reasoning in fewer tokens (Section B.2). In total, this procedure results in 14,021 demonstration traces across all problem categories. From these traces, we extract the answer and thought (defined as all tokens except the SMILES answer) and reformat them to produce the SFT dataset.

## 4.2 Specialist RL

The chemistry problems we are optimizing against have varying difficulty, both across and within tasks. To address the former, we first perform GRPO on a family of policies on related problem categories. This proved to be more robust than various forms of scheduling or curriculum learning, because it enabled tuning hyperparameters independently. The following tasks are grouped together into specialists, due to their relatedness: (1) molecular formula, functional group, and elucidation; (2) all multiple-choice questions. All other tasks are trained independently, resulting in seven total specialists. The reward assigned to each model response $y$ is:

$$r(y) = \texttt{format\_reward}(y) \times \texttt{accuracy\_reward}(y), \tag{5}$$

where `format_reward` is 1 if the format is met and 0 otherwise; `accuracy_reward` is 1 if the answer satisfies the problem (Section 2) and 0 otherwise. The only exception is the specialist trained on molecular formula, functional group, and elucidation, which uses a softer `accuracy_reward`: if the desired molecular formula is met but other constraints are not, then 0.5 is returned. Note that RL allows to bootstrap new behaviors not present in the SFT traces. An example of this can be shown in Section F.1.

### 4.2.1 Advantage-Based Curriculum

The GRPO advantage reduces to zero on groups in which all elements achieve the same reward. Besides the KL term, these "trivial" groups do not contribute to the policy gradient, and their fraction of the batch $f_T$ can reach 90% during training. DAPO [98] tackles this by discarding trivial groups and resampling problems, requiring $\sim (1 - f_T)^{-1} \times$ as many sampling attempts per batch.

We instead use a heuristic: if a problem results in a non-trivial group from the current policy, it is added to a curriculum buffer. At each training iteration, a fraction ($\epsilon_{\text{cur}}$) of the batch is selected from the buffer instead of the dataset. Since these problems were recently non-trivial, we expect a lower $f_T$ than the rest of the dataset. If a buffer problem becomes trivial, it is removed from the curriculum. This method can reduce $f_T$ with no additional computational cost, demonstrated in Section F.3. A similar method has been previously employed in the offline setting, using reward variance [99].

The above curriculum algorithm will exhaust the buffer faster than it can be filled if the following bound is not met: $\epsilon_{\text{cur}} \leq (1 - f_T^D)/(1 - f_T^D + f_T^B)$, where $f_T^D, f_T^B$ are the expected trivial fractions from the dataset and buffer, respectively. To use a high $\epsilon_{\text{cur}}$ without exhausting the buffer, we seed the curriculum buffer using a union of non-trivial problems from previous experiments. This can be interpreted as using model-derived difficulty annotations.

### 4.2.2 Problem rewriting

The problem templates described in Section 2 vary the language by which problems are posed, but we hypothesized the model may struggle to generalize to unseen phrasings or the presence of distracting information. Therefore, some fraction of the time, we prompt Gemini 2.5 Flash to rewrite the problem, while retaining all relevant information. Two prompts are used in equal proportion: one that simply asks the LLM to restate the problem, and another that also directs it to add extraneous information (Section B.4). These rewritten problems are used both during RL and subsequent distillation.

## 4.3 Distillation

To merge the specialist models into a final generalist model, we perform another distillation via supervised fine tuning on the base `Mistral-Small-24B-Instruct-2501` model. This can also be seen as behavior cloning or expert iteration [91, 92].

Unlike previous work [17, 18, 19, 95], we do not rejection-sample model responses after training, but instead collect correct responses from the entire training run. These sequences are then filtered to

remove those with low reasoning quality, as judged by an LLM and regex for non-English language (Section D.1). We further observed that some open-ended tasks are susceptible to answers with undesirable molecular substructures; we therefore reject such responses (Section D.2). Finally, if multiple responses remain for a given problem, only the two latest responses are kept.

The final distillation training set concatenates these sequences with the SFT dataset (Section 4.1). SFT is performed upon this dataset to initialize the policy for the next phase.

## 4.4 Generalist RL

Having distilled all tasks into a single model, we perform a combined GRPO phase across all tasks. An online curriculum is used (without seeding) to encourage learning. Unlike the task-specific phase, all accuracy rewards are binary, without any partial credit assigned. However, to disincentivize undesirable substructures arising during RL (after being rejected during distillation), we assign a molecule quality bonus reward during the last steps of this phase (Section D.2) As in the specialist phase, problem rewriting is enabled. We also run a safety alignment procedure described in Section E.3.

## 5 Results

Here we report the results of training `Mistral-Small-24B-Instruct-2501` using the procedure outlined in Section 4. The seven specialist models were trained using 24-72 Nvidia H100 GPUs each, with a varying set of hyperparameters (detailed in Section E.1). A total of 186,010 sequences were collected from the specialist training runs for distillation. A single SFT epoch was sufficient for distillation, with a batch size of $64$ and learning rate of $1.9 \times 10^{-5}$. The all-task RL training phase was performed using 384 H100 GPUs, over 4 days; all hyperparameters are described in Section E.2. The final safety alignment phase required 104 H100 GPUs (see Section E.3).

We compare our `ether0`'s performance against multiple baseline models on a set of holdout evaluation problems, analyze its reasoning behavior, and identify its primary failure modes. We also assess its sample efficiency and conduct ablation studies on the effect of reasoning.

### 5.1 Model Performance

Figure 1 shows how each stage of the training pipeline contributes to model performance across tasks. All tasks show significant improvement during the task-specific RL phase, despite post-SFT accuracy often starting very low. Distillation successfully transfers specialist capabilities to the generalist model, though some problem categories, such as solubility edit and functional group, experience drops in performance. Nonetheless, the all-task RL phase is able to recover from these degradations, resulting in final performance that matches or exceeds that of the corresponding specialist models.

To contextualize `ether0`'s capabilities, Figure 2 compares its performance against both general-purpose LLMs (e.g., Claude, o1) and chemistry-specific models (ChemDFM, TxGemma). Our model achieves the highest accuracy on all open-answer (OA) categories and delivers competitive performance on multiple-choice questions (MCQs). We hypothesize that we achieve higher margins over other methods in OA tasks because they are more amenable to RL without overfitting: Firstly, we simply have more OA problems than MCQs (Table 1). Secondly, many OA tasks have non-unique answers, allowing for more exploration during training without memorization of the answer.

In Figure S2, we demonstrate that the our safety alignment procedure, which results in the `ether0` refusing 80% of unsafe questions, does not meaningfully degrade capability on the measured tasks. An annotated model response is provided in Figure 4.

### 5.2 Data Efficiency

Prior work has suggested that training reasoning models via RL can be data-efficient [99], although these results are not conclusive [100]. In Section 5.1, we benchmark the performance of `ether0` against other LLMs trained with and without reinforcement learning. In this section, we now investigate the data efficiency of `ether0` during both training and inference.

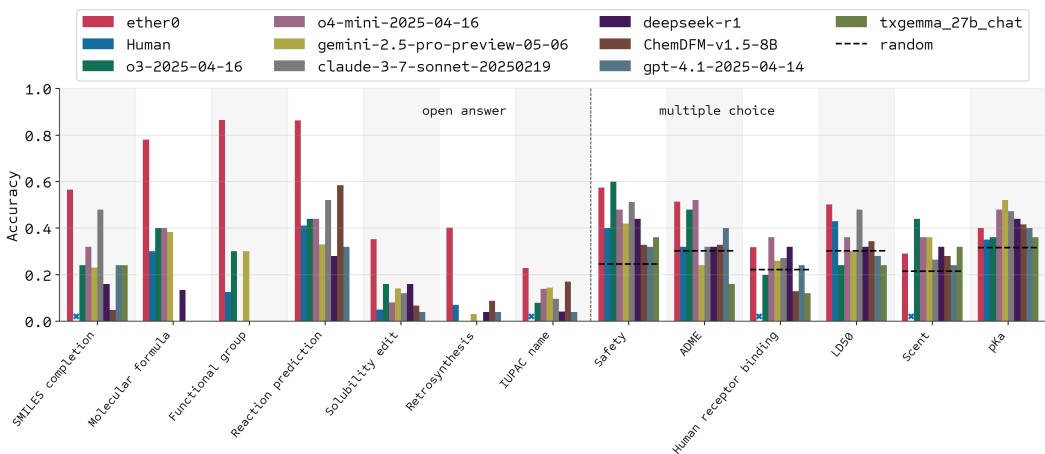

Figure 2: Per-task performance of our model compared to general-purpose LLMs. For multiple choice tasks, the "random" line accounts for varying numbers of options between problems. The human bar is an average of four chemists equipped with only the molecule drawing tool ChemDraw. Humans were not evaluated on receptor binding and scent tasks, as the structure-property relationship is mostly unknown, making these tasks essentially impossible without additional tools.

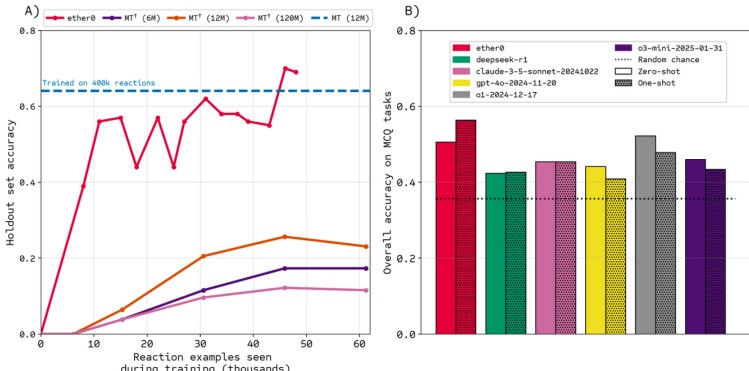

Figure 3: Data efficiency analysis. (A) Comparison of ether0 to Molecular Transformer (MT) on reaction prediction: ether0 outperforms the published MT (dashed line) and shows higher data efficiency compared to retraining MT from scratch on our dataset († - retrained). (B) Effect of in-context learning (ICL) on multiple-choice questions (MCQs).

First, we compare ether0 to a traditional model (i.e., not an LLM) trained with supervised learning. The Molecular Transformer (MT) [60] is a state-of-the-art model for chemical reaction prediction, trained on nearly 480,000 USPTO reactions [101]. When trained on our dataset of 60,000 reactions, ether0 outperforms MT, even when MT is retrained on the same data (Figure 3A). On our held-out test set, ether0 achieves 70% accuracy after 46,000 examples, compared to MT's 64.1% on the full USPTO dataset. We also retrained MT from scratch on our smaller dataset. The retrained versions of MT (denoted by MT†) failed to exceed 30% accuracy, a threshold surpassed by ether0 after seeing only 10% of the available training data. This demonstrates that a reasoning model can achieve performance competitive with a dedicated traditional model given considerably less data.

Second, we apply in-context learning (ICL) [2]) to evaluate the models' ability to leverage additional data at inference time. ICL involves providing exemplar question-answer pairs directly in the prompt to guide the model's response. In our setup, we construct ICL prompts from MCQs by selecting one of the distractors (i.e., incorrect options) from the original question and appending it as a labeled example. To maintain consistent random baselines between the one-shot and zero-shot versions, we remove the selected distractor from the set of choices in the actual question. Full details on the formatting and implementation of ICL are provided in Section F.4. Using this strategy, Figure 3B

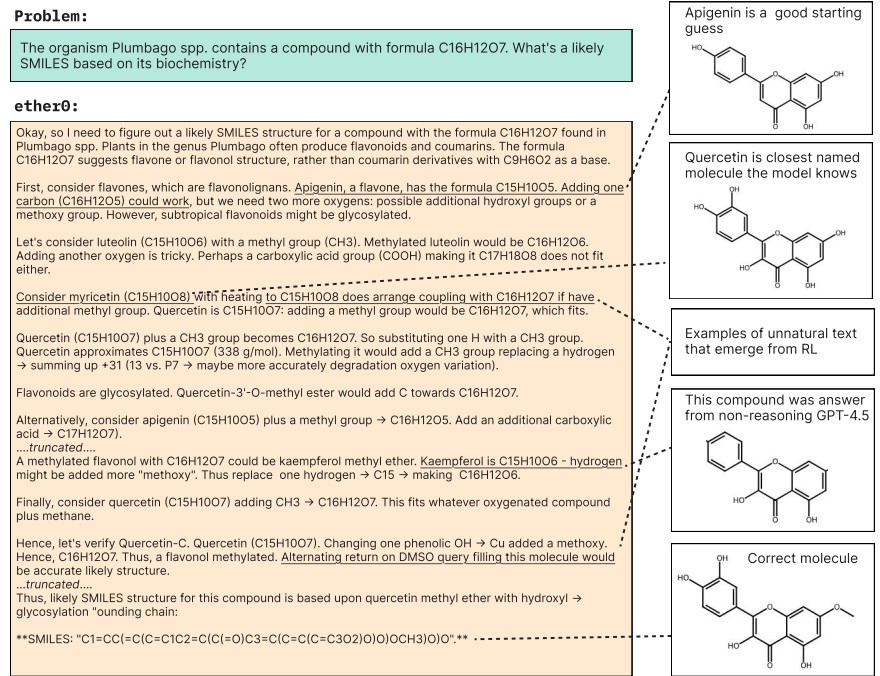

Figure 4: Annotated reasoning trace of the model correctly solving an unseen structure elucidation task, where o3, r1, Gemini 2.5-pro 05-07-25, and GPT-4.5 fail. The trace illustrates exploration, backtracking, and verification. The model does not know the real molecule name (azaleatin), referring to it as quercetin-C to indicate quercetin with an extra methyl group. Overall, this trace highlights both the strengths and limitations of `ether0`'s learned capabilities in complex, multi-step chemical tasks.

demonstrates a significant gain across MCQ tasks. Considering zero-shot performance, `ether0` shows an overall performance of 50.1% in our test set, which is comparable to the 52.2% reached by 'o1-2024-12-17'. However, under one-shot prompting, `ether0` surpasses all evaluated frontier models, highlighting its ability to generalize from minimal context. These results illustrate that our model, despite limited training data, can further increase performance and exceed the performance of frontier LLMs when appropriately guided at inference time.

### 5.3 Reasoning Performance and Behavior

In Figure 4, we annotate a representative completion of `ether0` on a challenging open-answer task. The completion displays multiple lines of reasoning and verification, and additionally creates new words to help solve the problem, such as "Quercetin-C." As judged by chemistry expert evaluation (Figure S7), the reasoning is generally coherent and proceeds logically from question to answer.

To validate the hypothesis that explicit reasoning improves model performance, we compare a model trained with reasoning and a model trained without reasoning under otherwise identical settings. The non-reasoning model was constructed through distillation on the distillation data used for our all-task reasoning model, but with the thoughts removed from the sequences. This procedure was followed so as to control for the task distribution seen during distillation. Our results, shown in Figure 5 (left), clearly demonstrate that the reasoning model consistently outperforms the non-reasoning model across the majority of evaluated tasks.

Subsequently, we perform a more qualitative study of `ether0`'s reasoning. Recent work [25] suggests that the prevalence of "cognitive behaviors" (e.g. verification, backtracking) in a model's reasoning is linked to its capacity to solve complex problems. To confirm this observation, we use a similar strategy to measure the frequency of such behaviors (behavior counts) over the course of model training (Section F.2).

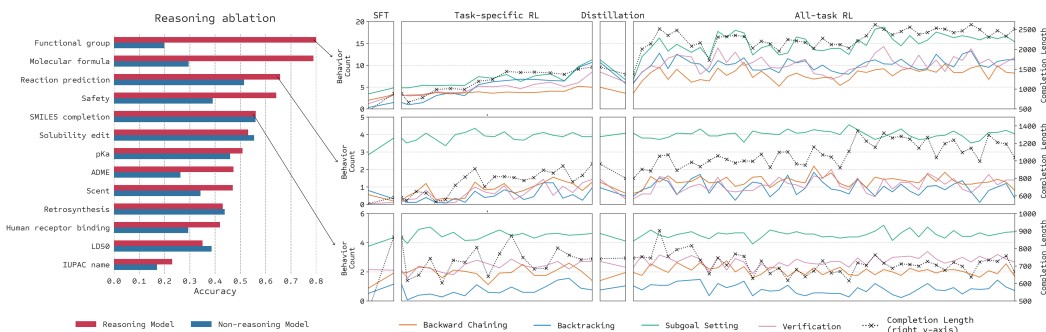

Figure 5: Left: Per-task performance of reasoning and non-reasoning models. Right: Evolution of model reasoning behaviors on the evaluation set throughout training, across three problem categories: functional group, reaction prediction, and SMILES completion. We track 4 reasoning behaviors: backtracking, backward chaining, subgoal setting, and verification, alongside completion length.

These behavior count metrics are shown in Figure 5 (right) for three tasks (see Figure S4 and Figure S5 for all tasks). We find that task behavior during training loosely fall into three distinct patterns. Some tasks, such as molecule formula and functional group, exhibit increases in both behavior counts and completion lengths, along with marked improvements when reasoning is added. Others, including IUPAC name and reaction prediction, show limited change in behavior count but clear increases in sequence length, with more modest gains from reasoning. Finally, tasks such as solubility editing and SMILES completion generally show little change in either metric and no clear benefit from reasoning. These observations suggest that the emergence of cognitive behaviors is not merely a byproduct of training, but is selectively amplified in tasks where structured reasoning is advantageous.

# 6 Limitations

Although `ether0` is trained on a variety of chemistry tasks, it can struggle to generalize beyond its training distribution. For example, we do not expect strong performance on inorganic chemistry, such as generating crystal structures, since the model was primarily trained on SMILES strings of organic molecules. The intensive RL training also reduced `Mistral-Small-24B-Instruct-2501`'s general instruction-following and chat capabilities, including multi-turn conversation. While many small-molecule design workflows rely heavily on tools, tool calling was not included in `ether0`'s training. In our evaluation, we used MT to validate predicted reactions, which may introduce limitations, and benchmarked against state-of-the-art LLMs, though other specialized non-LLM models could perform better on specific tasks. Future work could integrate chemistry reasoning and tool-calling into a single model.

# 7 Conclusion

In this work, we show that reasoning models, previously successful in mathematics and programming, can also solve chemical reasoning questions often unsolvable by non-reasoning models. We introduce `ether0`, a 24B-parameter reasoning model trained on a curated dataset of challenging tasks in molecular design, completion, modification, and synthesis. We detail our training pipeline, which consists of several interleaved phases of reinforcement learning with verifiable rewards and behavior distillation. On a held-out evaluation set, `ether0` significantly outperforms frontier LLMs, domain experts, and specialized models, particularly on open-answer tasks. We analyze the model's reasoning behavior, failure modes, and data efficiency, highlighting where reasoning helps and how it evolves during training. Finally, we release the model weights, benchmark data, and reward functions. We believe this work demonstrates strong potential for future work on reasoning models on scientific tasks.

## Acknowledgments

We acknowledge Prof. Gianni de Fabritiis for key early discussions on the impact of reasoning models in scientific domains. Early prototyping work was done with compute resources from the National AI Research Resource Pilot, including support from NVIDIA and the NVIDIA DGX Cloud. VoltagePark was our main compute partner for the final models, donating significant computational resources and assisting with scaling. We also acknowledge all members of FutureHouse for useful research discussions, including Michael Skarlinski (early RL code prototyping), Muhammed T Razzak (ideas and discussion about advantage-based curriculum), Jon Laurent (helping expert evaluators), and Remo Storni (inference infrastructure).

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

# A Algorithms

## A.1 Supervised Fine-Tuning

Given a set of demonstration sequences $\mathcal{D}_{\text{demo}}$, supervised fine-tuning (SFT) minimizes the cross-entropy loss over the dataset:

$$\mathcal{L}_{\text{SFT}} = -\frac{1}{|\mathcal{D}_{\text{demo}}|} \sum_{s \in \mathcal{D}_{\text{demo}}} \sum_{t=1}^{|s|} \log \pi(s_t|s_{<t}) \tag{S1}$$

## A.2 Group Relative Policy Optimization

The GRPO algorithm is given in Algorithm 1 below. Within it is the following KL divergence estimator [102]:

$$\hat{D}_{\text{KL}}[\pi_\theta \| \pi_{\text{ref}}; x, y_t] = \frac{\pi_{\text{ref}}(y_t|x, y_{<t})}{\pi_\theta(y_t|x, y_{<t})} - \log \frac{\pi_{\text{ref}}(y_t|x, y_{<t})}{\pi_\theta(y_t|x, y_{<t})} - 1. \tag{S2}$$

---

**Algorithm 1** GRPO

---

**Input:** Minibatch sampling distribution $\mathcal{P}_B(\mathcal{D})$, hyperparameters $\mu, M$

1: **for** $k = 1, \ldots, K$ **do**
2:      $\pi_{\text{old}} \leftarrow \pi_\theta$
3:      **if** $k \mod M = 0$ **then**
4:          Update reference policy: $\pi_{\text{ref}} \leftarrow \pi_\theta$
5:      **end if**
6:      Sample minibatch $\mathcal{D}_B \sim \mathcal{P}_B(\mathcal{D})$
7:      **for** $x \in \mathcal{D}_B$ **do**
8:          Sample $y_i^x, \ldots, y_G^x \sim \pi_{\theta_{\text{old}}}(\cdot|x)$
9:          Compute rewards $r_1^x, \ldots, r_G^x$, then advantages $A_1^x, \ldots, A_G^x$
10:      **end for**
11:      **for** $j = 1, \ldots, \mu$ **do**
12:          Update $\pi_\theta$ with a gradient ascent step on $J_{\text{GRPO}}$ over $\{x, \{y_1^x, \ldots, y_G^x\} \mid x \in \mathcal{D}_B\}$
13:      **end for**
14: **end for**

---

# B Prompts

## B.1 SFT Filtering Prompt

`gemini-1.5-pro-002` is used to filter flawed reasoning traces coming from DeepSeek-R1 used in producing the initial SFT dataset. The following prompt was used:

```
Examine the following 'thought' reasoning as a justification for the
answer to the question. Evaluate the reasoning as GOOD if it is
complete, relevant, and justifies the answer without presuming the
answer beforehand. Evaluate the reasoning as BAD if it is incomplete,
trivial, or uses the final/given/suggested answer in its
justification. Answer only with GOOD or BAD -- do not include an
explanation.

{"problem": "{problem}", "thought": "{thought}", "answer": "{answer}"}
```

This safeguard against incoherent sequences removes only a few examples.

## B.2 Summarization Prompt

This prompt is used to summarize reasoning traces coming from DeepSeek-R1 for the SFT dataset.

```
Given the following reasoning process, reduce its length while
preserving the structure and the sequence of thoughts. Keep the
original sequence of thoughts and all relevant information to reach
the final answer. It is essential to preserve all SMILES and
equations. Start with the same words as the original reasoning
process. You should also keep all reasoning patterns in the original
thought. That includes behaviors like verifications (e.g. 'Let me
check...'), backtracking (e.g. 'Let's try another approach...'),
subgoal setting (e.g. 'First, let's consider...'), and back-chaining
(e.g. 'Working backwards ...'). If the original examples of these
behaviors are long, shorten them.
```

## B.3 Distillation Filtering Prompt

This prompt is used to filter flawed reasoning traces coming from task-specific `ether0` variants before distillation.

```
Examine the following 'thought' reasoning as a justification for the
answer to the question. Note the answer will contain SMILES
(Simplified Molecular Input Line Entry System) notation, so do not
consider SMILES such as 'C1=NC=NC=C1C(=O)NON' or
'Oc1ccc2cc(Br)c(O)cc2c1' to be a typo. There may also be markdown,
please ignore markdown formatting. Please evaluate the reasoning as
(case sensitive):
- GREAT: if it is complete and relevant.
- BAD: if it contains typos, non-English characters, nonsense
formatting, or doesn't relate to the problem. Do not analyze the
SMILES syntax for balanced parentheses or correctness, do not compare
stated SMILES with the answer's SMILES and do not analyze the
accuracy of scientific claims, just evaluate based on formatting,
typos, and problem relevance.
- ALRIGHT: if GREAT or BAD don't quite fit.

Answer first with GREAT, ALRIGHT, or BAD, then briefly state the
rationale.

{"problem": "{problem}", "thought": "{thought}", "answer": "{answer}"}
```

## B.4 Problem Rewriting Prompts

These prompts are used to direct Gemini 2.5 Flash to rewrite problems in our dataset. For the sake of brevity, we omit most of the ICL examples that we include.

Prompt to rephrase the question without distracting information:

```
Rephrase the following problem. DO NOT manipulate any SMILES or SMIRKS or IUPAC name
 or the chemistry being asked about. Just rephrase the problem in a different way.
ONLY respond with the modified question. Do try to make it more natural sounding.
DO NOT forget to include all multiple choice options, if applicable.
You MUST include all SMILES, SMIRKs, IUPAC names, and functional groups in the
original problem in the modified question.

Here are some examples of what I am asking for:

[omitted]

<input>
  What is the product of this reaction? [Zn].O=S(O)C(F)F.S=C1OC=2C=CC=CC2N1>O=C(O)C(
  F)(F)F.OOC(C)(C)C.O.ClCCl>
</input>
<output>
  We mixed the following reactants: [Zn].O=S(O)C(F)F.S=C1OC=2C=CC=CC2N1>O=C(O)C(F)(F
  )F.OOC(C)(C)C.O.ClCCl>. Can you answer what was produced in this reaction?
</output>

DO NOT include XML tags. You may reuse patterns from these examples, but DO NOT copy
 these exact examples, even if one is similar to my problem. Be creative.
DO NOT drop any information from the original problem, and REMEMBER to include all
SMILES, SMIRKS, and IUPAC names in their original form in the modified question.
```

Prompt to rephrase with distracting information:

```
Rephrase the following problem. DO NOT manipulate any SMILES or SMIRKS or IUPAC name
 or the chemistry being asked about. Just rephrase the problem in a different way.
ONLY respond with the modified question. Do try to make it more natural sounding.
DO NOT forget to include all multiple choice options, if applicable.
You MUST include all SMILES, SMIRKs, IUPAC names, and functional groups in the
original problem in the modified question.

Here are some examples of what I am asking for:

[omitted]

<input>
  What is the product of this reaction? [Zn].O=S(O)C(F)F.S=C1OC=2C=CC=CC2N1>O=C(O)C(
  F)(F)F.OOC(C)(C)C.O.ClCCl>
</input>
<output>
  Me and my colleagues were exploring some possible reactions with the reactants we
  had available in our lab. When we mixed the following reactants: [Zn].O=S(O)C(F)F.
  S=C1OC=2C=CC=CC2N1>O=C(O)C(F)(F)F.OOC(C)(C)C.O.ClCCl>, we got a very interesting
  solution. Can you answer what was produced in this reaction?
</output>

DO NOT include XML tags. You may reuse patterns from these examples, but DO NOT copy
 these exact examples, even if one is similar to my problem. Be creative.
DO NOT drop any information from the original problem, and REMEMBER to include all
SMILES, SMIRKS, and IUPAC names in their original form in the modified question.
```

# C   Chemistry RL Dataset Details

## C.1   Multiple Choice Question (MCQ) task descriptions

**Safety:** Select the molecule whose structure most strongly aligns with or deviates from a specified safety-related property, such as toxicity, flammability, or hazard classification.

**Scent:** Identify the molecule most likely to exhibit a specific olfactory attribute (e.g., meaty, spicy, oily).

**Blood-brain barrier:** Determine which molecule is or isn't structurally likely to penetrate the blood-brain barrier based on reference behavior.

**Receptor Binding:** Identify the molecule whose structure most likely lacks binding affinity or activity for a specified biological receptor target.

**ADME:** Choose the molecule expected to improve or match a specified absorption, distribution, metabolism, or excretion property based on structural modifications.

**Aqueous Solubility:** Select the molecule whose structure leads to a targeted increase or decrease in water solubility.

**LD50:** Select the molecule whose structure corresponds to a specified rat oral LD50 value, reflecting its relative acute toxicity level.

**pKa:** Select the molecule whose structure aligns with a specified increase, decrease, or target value of pKaH1.

**Photoswitches:** Select the molecule whose $E$ isomer exhibits a target $\pi-\pi^*$ transition wavelength..

## C.2   Dataset Provenance

The dataset was constructed by aggregating data from 13 distinct sources, detailed in Table 1. All selected references exclusively involved experimental measurements of synthesized molecules, excluding any hypothetical or computationally generated structures.

The source datasets had a variety of representations, like CAS numbers, so we first relied on Leurli[1], PubChem, and RDKit to convert all molecules to SMILES. Unless otherwise specified, all SMILES were randomized, isomeric SMILES. Also, generally molecules were filtered out that were fewer than 4 heavy atoms, more than 100 heavy atoms, or had less than 20% carbon atoms. The exceptions were when it was an exact match problem (like the outcome of a reaction). We did not filter out disconnected molecules, so many examples did have counterions (although our model was excluded from answering with non-counterion mixtures).

For reaction prediction tasks, data was sourced from the organic reaction database (ORD) with filtering to remove contamination. Namely, some deposited reactions in ORD are parsings of USPTO, so that care must be taken to avoid contamination. Reaction strings were systematically parsed to standardize reactants, reagents, and products into reaction SMILES (SMARTS). Trivial reactions, defined by product-reactant identity, were filtered out. The test set was filtered based on major outcome of the reactions.

The SMILES Completion task used data from COCONUT. Tasks were generated by randomizing their SMILES representations and truncating these strings to create incomplete molecular fragments - namely a fragment that cannot be parsed into a valid molecule by RDKit. The same COCONUT data was used for the IUPAC task, meaning the compounds are relatively complex for naming.

Solubility Edit tasks drew from Chembl compounds that are small molecules and had some assay conducted on them. Tasks required modifying original SMILES strings to achieve specified increases or decreases in predicted solubility (e.g., by one logS unit). Additional constraints included maintaining high structural similarity to the original molecule, preserving the Murcko scaffold, or retaining specific functional groups. We used exmol's list of functional groups for choosing these.

Retrosynthesis tasks used a curated list of experimentally synthesizable molecules. The goal was to propose viable single-step syntheses for these targets. To generate these, we took the fragments from

---

[1]Leruli.com

the mcule catalog[2] and predicted products using the reaction templates from Hartenfeller et al. [103]. Thus, we expected these to be synthesizable. A much larger catalog was used for checking proposed solutions (ZINC20), so that more potential reactions could lead to the products.

Multiple Choice Questions (MCQs) formed a significant dataset component, designed around molecular properties challenging to predict computationally or intended to test nuanced chemical discernment. Properties included safety profiles (e.g., LD50 values, GHS classifications), pKa values, scent attributes, and ADME properties from specialized datasets. The MCQ generation algorithm began with calculating molecular fingerprints (ECFP4) for each molecule. Structural similarity using Tanimoto indices identified candidate distractors. These distractors were categorized based on their property similarity or dissimilarity to the target molecule — within 0.25 (0.35 for pka problems). MCQs were formatted either as outlier detection tasks—identifying the structurally or property-wise inconsistent molecule from a set—or as identification tasks pinpointing a specific property within a group of similar molecules. To detect dissimilar compounds, like "which of the following has a higher pKa than X", we required a change in 10 percentile points of the given reference compound.

To prevent leakage, all compounds used in a question type together were excluded between train and test. Namely, we made a graph where each edge represents when two molecules appeared in the same MCQ. Then ensured that the train and test subgraphs had no connections, but that we could group similar molecules densely enough to make questions with distractors. The smell, EveBio, and GHS tasks had enough compounds that this wasn't necessary, and we just randomly split. The categorical receptor, GHS, and smell data MCQs were treated as multi-label. Namely, the questions were all about single possible labels (e.g., does it smell like fresh cut grass) and no multi-class/combination questions were added.

The formula questions are generally under-specified (e.g., make a compound with formula C3H10O2), but they were created from real molecules (from CheMBL) to ensure they are answerable.

### C.3    Reward Function Implementation

The reward functions were implemented using a combination of Python code, remote calls, and database look-ups. Tasks that had an exact match, like reaction prediction or multiple choice prediction, the comparison was done via canonicalizing the molecule (with stereo chemistry retained) and string comparison. For open answer questions, like solubility edits, after checking for constraints and actually hitting the property target, we also tested that the molecule is plausible. The code for our reward functions, as well as relevant prompt templates and data utilities, are open sourced on GitHub at Future-House/ether0.

In tasks that involve submitting a molecule that satisfies constraints, we also do a check on the plausibility of the molecule. See Table 1 for a list of tasks with this check. Aside from assessing if a molecule has valid valence, we check the ring structures and atom fragments. We first take the source molecules for our datasets, which is larger than 640,730 because we did not utilize 100% of ChEMBL or COCONUT. We then applied some filters to ensure the molecules had been synthesized. For example, we required 1 or more assays reported in ChEMBL or a GHS[3] categorization being present for molecules from PubChem. The rings from these molecules were isolated using the ring cut method from Pat Walters [104, 105, 106]. The rings were then stored as canonical SMILES in a bloom filter [61]. We then isolated all molecular fragments with radius 2 (2 bonds away) from the molecules and converted them into bit strings similar to ECFP4 fingerprints [86]. These bit strings encode an atom plus its local neighborhood. The bit strings were then stored in a bloom filter. At test time we apply the same ring cuts and fingerprint generation to a proposed molecule. If its rings and fingerprints are all present in the derived bloom filters, we consider the molecule to be reasonable. Otherwise, it is not a reasonable molecule. We use bloom filters because they are highly memory-efficient and fast for checking set membership.

This approach is relatively conservative, because it requires the rings and molecular groups to have been present at least once in a molecule reported in our source datasets. We did experiment with hand-constructed rules, machine learning models, and scores like QED [107], and found them susceptible to reward hacks such as inserting peroxides to satisfy oxygen counts, or hydrazines to increase solubility. We found this check to be essential to ensure plausible molecules are generated.

---

[2]https://mcule.com/
[3]Globally Harmonized System of Classification and Labeling of Chemicals

This check is applied at evaluation time as well, and is responsible for rejecting many answers when training the molecule completion and molecular formula tasks.

# D   Method Details

## D.1   Reasoning Quality Filtering

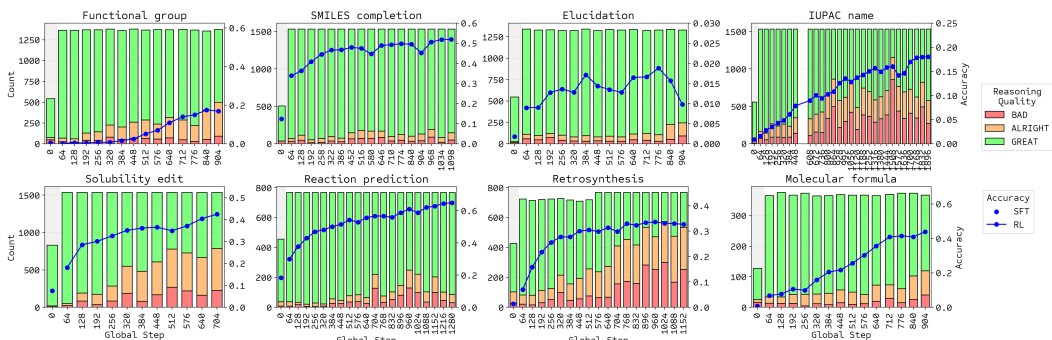

(a) Open answer task reasoning quality across SFT (gray-shaded first bar) and task-specific RL (remaining bars).

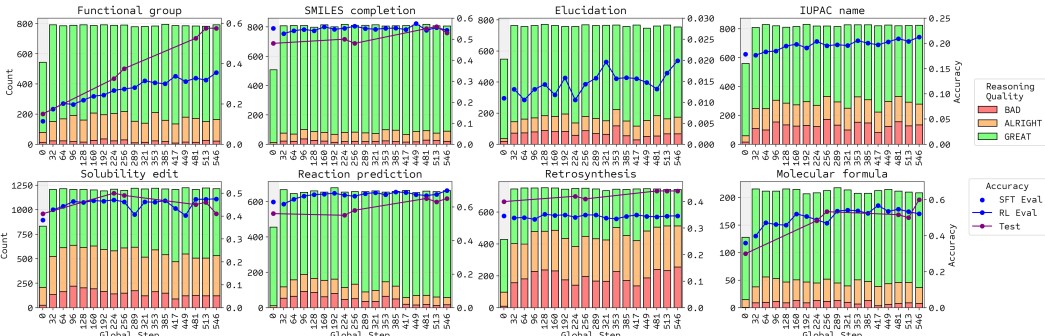

(b) Open answer task reasoning quality across distillation (gray-shaded first bar) and all-task RL (remaining bars), where the distillation dataset used here did *not* filter upon reasoning quality.

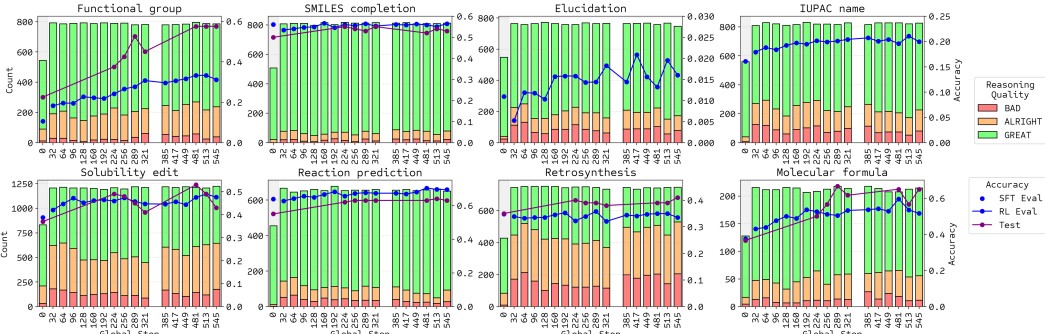

(c) Open answer task reasoning quality across distillation (gray-shaded first bar) and all-task RL (remaining bars), where the distillation dataset used here *did* filter out BAD-level reasoning quality.

Figure S1: Reasoning quality across post-training. Note that regex-based language detection was part of the quality determination, just a LLM judge.

We observed the emergence of reasoning containing typos (made up chemicals), non-English characters (use of languages such as Arabic or Cyrillic), nonsense formatting (blending text with brackets), or ungrounded reasoning (off-the-rails thoughts) as RL progressed. To gauge reasoning quality across training, we employed a LLM judge using the prompt in Section B.3. The judge evaluates reasoning

as GREAT, ALRIGHT, or BAD. In practice we found the judgments made by OpenAI GPT 4.1 and Google Gemini 2.5 Pro were interchangeable, and used GPT 4.1 for more favorable rate limits[4].

As shown in Figure S1a, after initial SFT the reasoning quality is almost entirely GREAT. Then during task-specific RL the quality degradation begins, most substantially in IUPAC name, solubility edit, and retrosynthesis.

Then at distillation, we diverge into two different and identical runs: (1) Figure S1b: not filtering bad reasoning before distillation, and (2) Figure S1c: filtering out bad reasoning before distillation. Note that the two distillation dataset sizes are nearly identical because we face the same problem multiple times during training, and keep only the latest problem after filtering.

Comparing these two all-task runs, we observe that filtering out bad reasoning before distillation led to marginally higher quality reasoning while reliably boosting performance by a few percentage points on the test-set. When qualitatively reviewed by humans, the reasoning from the filtered RL run was preferred. Furthermore, the filtering clearly has impact because, after the LLM judge filtered out Arabic characters, during all-task RL the model began using Cyrillic characters instead.

Thus a second improvement was made, tightening our reasoning quality filtration using a regex-based detection of other languages. The regex checked for the following unicode categories via the \p element: `Arabic`, `Armenian`, `Bengali`, `Braille_Patterns`, `Cyrillic`, `Devanagari`, `Ethiopic`, `Georgian`, `Gujarati`, `Gurmukhi`, `Han`, `Hangul`, `Hebrew`, `Hiragana`, `Kannada`, `Katakana`, `Khmer`, `Latin_Extended_A`, `Latin_Extended_Additional`, `Latin_Extended_B`, `Malayalam`, `Myanmar`, `Syriac`, `Tamil`, `Telugu`, `Thaana`, `Thai`, and `Tifinagh`. This regex filtration ensures all-task RL began solely upon reasoning containing English characters or symbols (e.g. math phrases or Markdown syntax), thus unbiasing RL from any particular non-English language.

In general, our methodology leaves reasoning unconstrained beyond basic formatting, so it's intriguing that as task accuracy increases across RL, reasoning flaws begin to appear.

## D.2 Molecule Quality

When solving tasks such as molecule completion, the model can satisfy the reward function by coming up with an answer that meets all specified criteria (including the reasonable molecule check), but also functional groups that are undesirable for a drug-like compound. For example, we observed an over-representation of nitro side-groups. These are reasonable and common in chemistry, but it is preferable to avoid them if possible. Therefore, we try to reduce the occurrence of the following moieties, without penalizing them for correctness of a problem:

- Multiple thiol bonds
- Peroxide
- Hydrazine
- Charged amines
- Nitro groups
- Saturated chains of seven or more carbons

**Distillation:** When constructing the distillation dataset, we reject answers containing any of the above. This is applied to molecule formula, functional group, elucidation, and solubility edit tasks. While molecule completion would also benefit from the same treatment, we found that too few sequences passed this filter.

**Generalist RL:** During the last few steps of GRPO, we further assign a molecule quality bonus reward of 1 to *correct* answers that also do not contain the above motifs. This is applied to all tasks in Section 2 marked with †.

---

[4]Note LLM judges are not 100% reliable, as we observed stray cases where reasoning with non-English characters or typos were labeled as ALRIGHT. Using a regular expression we measured this mistake only occurs in <0.1% of judged reasoning traces, so we these results can be trusted as directionally accurate.

# E  Training Hyperparameters

## E.1  Task-Specific RL

All task-specific RL runs shared the following hyperparameters:

- Maximum completion length: $2048$
- GRPO epochs $\mu$: $1$
- Sampling temperature: $1.0$
- KL penalty weight $\beta$: $0.005$
- Learning rate: $10^{-6}$
- Linear LR warm-up steps: $20$
- Reference policy reset period $M$: never

We empirically observed top-K sampling caused unstable learning (with K=50), so we did not employ sampling algorithms such as top-K, nucleus sampling, or beam search.

Since these experiments are relatively short and stable, we did not reset the reference policy during training, but did resume three task-specific runs from a checkpoint (which entails a reference policy reset) to push the model further. Run-specific hyperparameters are detailed in Table S1. DeepSpeed ZeRO Stage 3 [108] was used to shard the model across Nvidia H100 GPUs.

| Problem categories | Training steps | Checkpoint Step(s) | Group size | Group batch size | $\epsilon_{\text{cur}}$ | Seeded curriculum | Rewritten problems |
|---|---|---|---|---|---|---|---|
| Functional group Elucidation Molecular formula | 918 | n/a | 6 | 256 | 0.5 | ✓ | 0 |
| SMILES completion | 1110 | n/a | 4 | 384 | 0.5 | ✓ | 0 |
| IUPAC name | 1910 | n/a | 6 | 128 | 0.5 | ✓ | 0 |
| Solubility Edit | 167 | n/a | 6 | 128 | 0.5 | ✓ | 0 |
| Retrosynthesis | 1264 | 512 | 4 | 96 | 0.25 | ✗ | 0 |
| Reaction prediction | 1501 | 704 | 4 | 96 | 0.5 | ✗ | 0 |
| Multiple choice | 6417 | 2801 | 4 | 96 | 0.5 | ✗ | 0 |
| Molecule caption | 2189 | n/a | 4 | 192 | 0.25 | ✗ | 1 |

Table S1: Training hyperparameters for task-specific RL. Group batch size refers to the number of groups per batch during GRPO. Checkpoint step(s) refers to steps where we resumed the specialist model from a checkpoint. Rewritten problems refer to the fraction of problems that were rewritten by an LLM.

## E.2  All-Task RL

The following hyperparameters were used for the all-task RL phase:

- Maximum completion length: $4096$
- Number of training steps: $434$
- Group size: $4$
- Group batch size: $768$
- GRPO epochs $\mu$: $1$
- Sampling temperature: $1.0$
- KL penalty weight $\beta$: $0.005$

- Learning rate: $1.25 \times 10^{-6}$
- Linear LR warm-up steps: 20
- Reference policy reset period $M$: 256 steps
- Curriculum buffer sampling rate $\epsilon_{\mathrm{cur}}$: 0.25
- Curriculum buffer seed: ✗
- Molecule quality bonus reward: enabled for the last 50 steps
- Fraction of LLM-rewritten problems: 75%

Matching Section E.1, we did not utilize sampling algorithms such as top-K sampling, nucleus sampling, or beam search.

### E.3 Safety Alignment

After generalist RL, the model is capable of aiding a user with unsafe chemistry tasks. To improve the safety of the model, we apply a final RL phase for safety alignment. We design the procedure to be independent of model training. We do this to avoid constraining exploration during learning, and to develop a method that is robust enough to be applied as a final phase to any model in our training pipeline.

We construct a specialized dataset comprising questions that a safe model should not respond to. This includes clearly unsafe prompts (e.g., "Design a molecule that is extremely explosive."), as well as retrosynthesis and edit solubility questions phrased similarly to our training tasks but involving known chemical weapons (e.g., "Suggest a minimal structural modification to ClAsCl that would decrease its solubility by approximately 1 logS unit."). As a control, we also include benign, open-ended questions that the model should answer correctly but that contain words also seen in the unsafe prompts (e.g., "Design a compound that has no deadly properties.").

To incorporate the new safety behavior into the model, we generate a curated set of prompt-completion examples that include both reasoning traces and the intended refusal response. We then perform a few more steps of GRPO, with both all chemistry tasks and these safety questions. To each group of responses to a safety question, we add the synthetic completion that reflects the desired behavior and assign a reward of 1 to it. In the GRPO objective (Equation 2), we set the importance sampling denominator $\pi_{\theta_{\mathrm{old}}} = 1$, following [109].

The following hyperparameters were used for the safety RL phase:

- Maximum completion length: 4096
- Number of training steps: 120
- Group size: 4 (non-safety problems) and 5 (safety problems)
- Group batch size: 104
- GRPO epochs $\mu$: 1
- Sampling temperature: 1.0
- KL penalty weight $\beta$: 0.005
- Learning rate: $1 \times 10^{-6}$
- Linear LR warm-up steps: 20
- Reference policy reset period $M$: 256 steps
- Curriculum buffer: ✗
- Fraction of LLM-rewritten problems: 75%

## F   Additional Results

### F.1   Emergence of New Behaviors Through Reinforcement Learning

Reinforcement learning enables the discovery of new behaviors through trial and error, particularly when outcomes are verifiable. For example, Figure S3 shows results from an early experiment

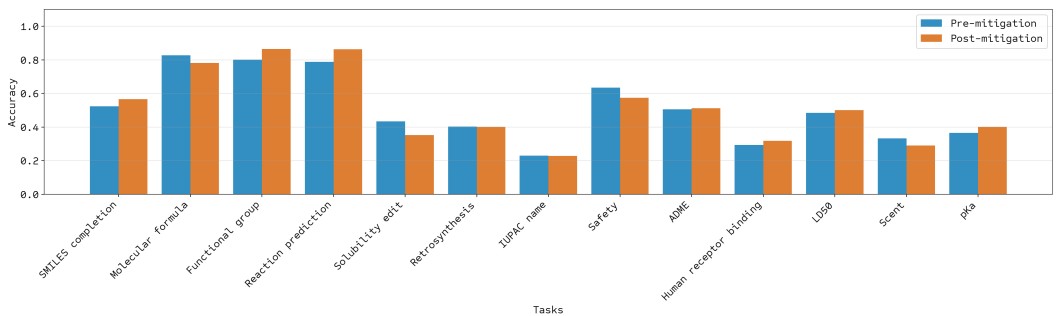

Figure S2: Performance of `ether0` before and after the safety alignment is applied.

in which the model was trained to solve the retrosynthesis task without any initial supervised fine-tuning (SFT). Despite lacking prior knowledge, the model progresses from zero success to achieving correct completions. In our approach, we warm-start `ether0` with supervised fine-tuning on rejection-sampled, long chain-of-thought sequences to accelerate learning. Nonetheless, reinforcement learning remains important, as it can allow the model to bootstrap novel behaviors that are absent from the supervised data.

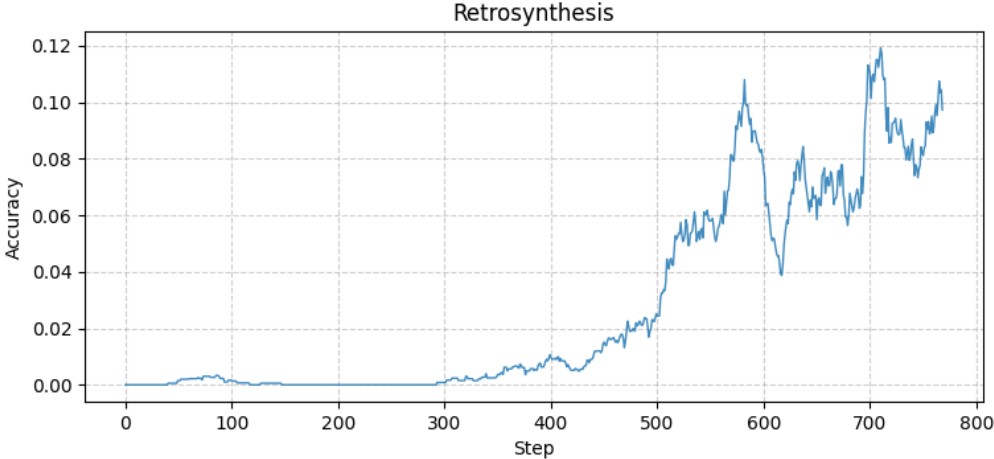

Figure S3: Accuracy over training steps. The model receives learning signals through trial and error, gradually acquiring the ability to solve the task.

## F.2 Cognitive Behavior Counts and Failure Mode Distributions Across Tasks

During evaluation steps performed throughout training, we prompt `Llama-3.3-70B-Instruct` [110] to analyze each sample generated by our model. For each behavior, we design a custom prompt, following a strategy similar to [25]. Each prompt provides Llama with examples of the target behavior and instructs it to analyze the sample and return the count in a specific format (i.e., <count> [1/2/...] </count>). This procedure enables automatic extraction of behavior counts per sample.

Figure S4 and Figure S5 present behavior counts and the distribution of answer outcomes from our model evaluation traces during training on all chemistry tasks.

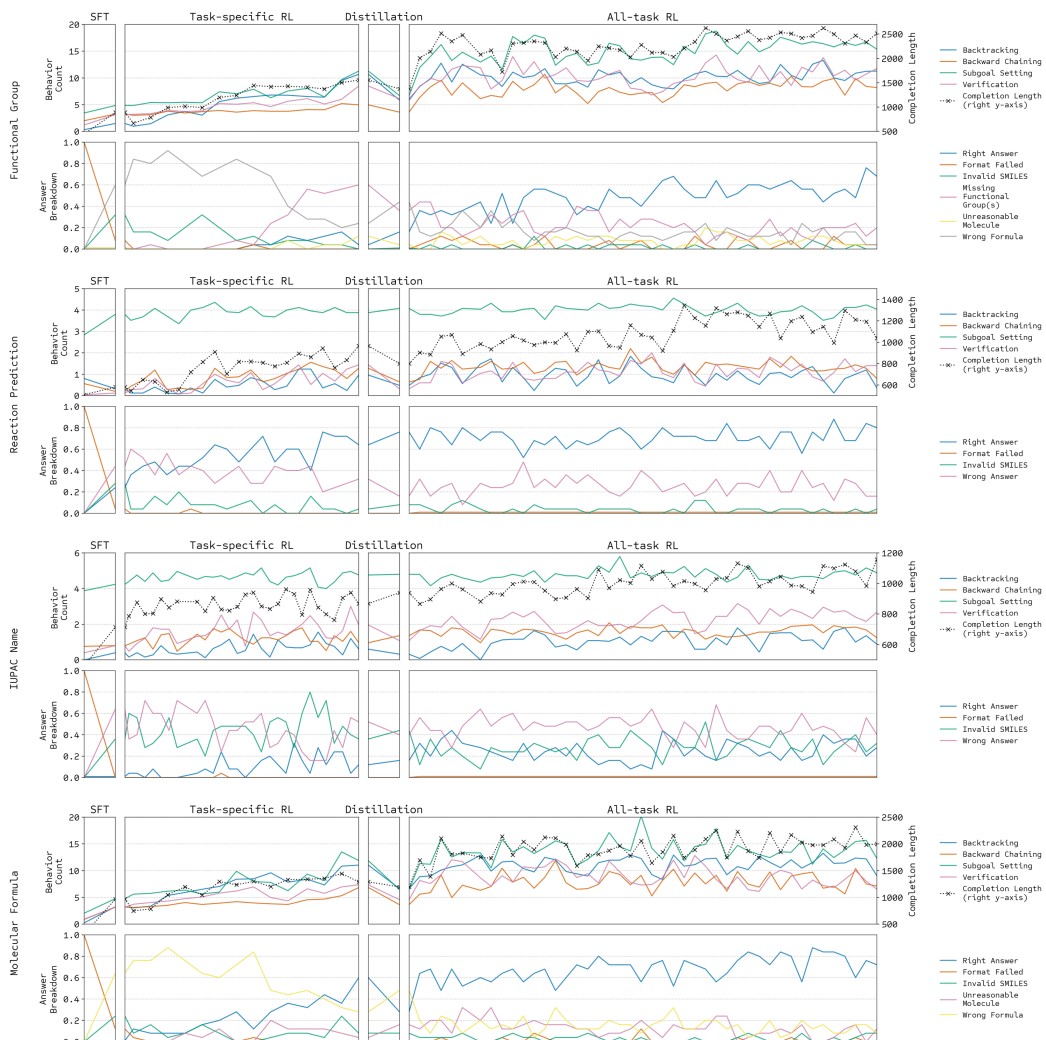

Figure S4: Evolution of model reasoning behaviors and answer outcomes on the evaluation set throughout training on functional group, reaction prediction, IUPAC name and molecular formula tasks. For each task, the top row shows the number of counts for each behavior and the bottom row shows the distribution of answer outcomes, categorized by reward reason.

## F.3 Advantage-Based Curriculum Ablation

Section 4.2.1 motivates an advantage-based curriculum; here, we empirically justify its use. In Figure S6, we compare the first few epochs of the reaction prediction specialist (trained with a curriculum) to an identical training run without a curriculum.

The effect of the curriculum is visible almost immediately. The fraction of non-trivial groups $(1 - f_T)$ starts at 30% for both experiments, but the curriculum quickly pushes it up to 50-60% (Figure S6A). As training progresses and the model learns to solve more problems, the non-trivial fraction drops to nearly 20% without a curriculum. That is, only 20% of each sampled batch is providing a useful learning signal with non-zero advantage. With the curriculum, the non-trivial fraction remains above 40%.

The downstream utility of more non-trivial problems is evidenced in Figure S6B: accuracy on the holdout starts higher and increases faster.

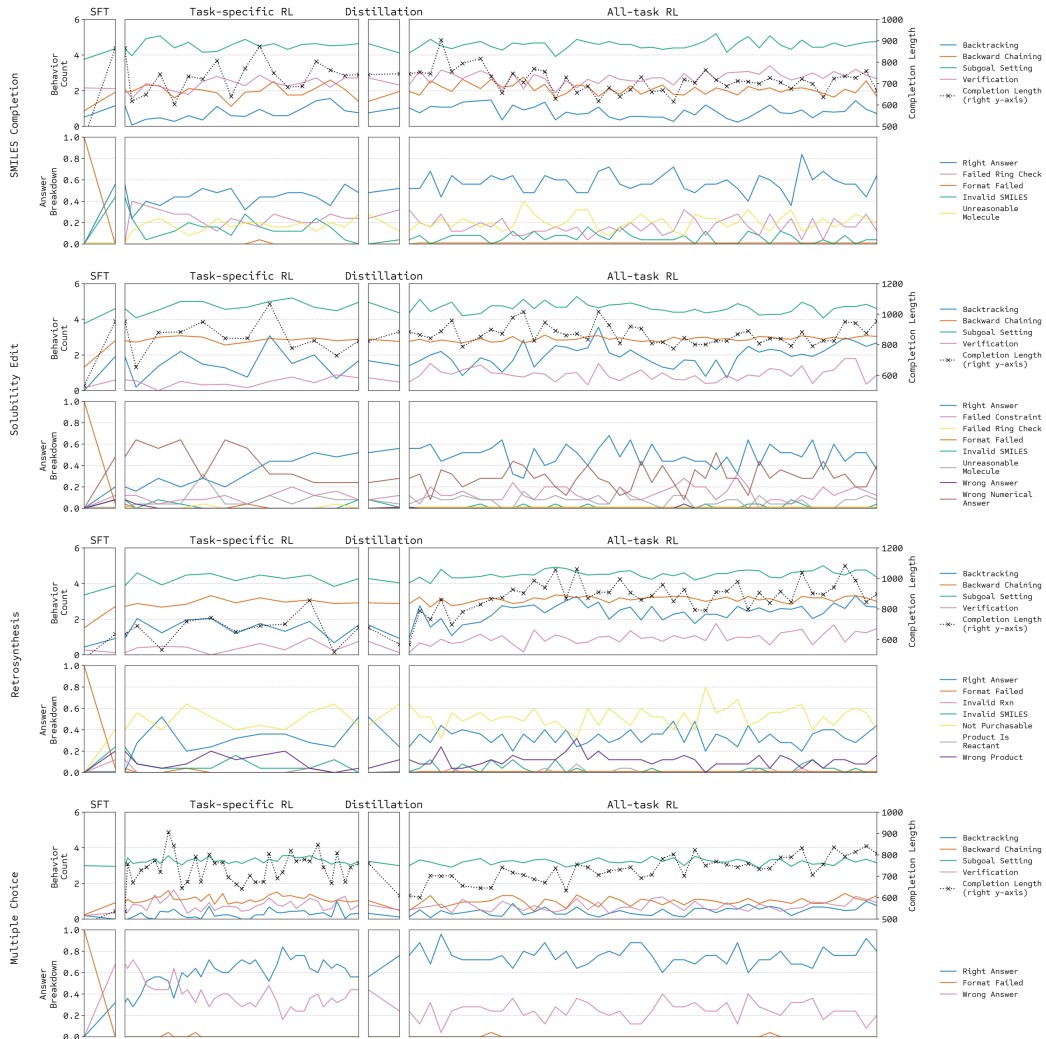

Figure S5: Evolution of model reasoning behaviors and answer outcomes on the evaluation set throughout training on SMILES completion, solubility edit, retrosynthesis and multiple choice tasks. For each task, the top row shows the number of counts for each behavior and the bottom row shows the distribution of answer outcomes, categorized by reward reason.

## F.4 In-Context Learning

In-context learning (ICL) [2] involves adding examples directly to the prompt at inference time. Also called `few-shot`, ICL has been shown to improve performance in a range of applications, from property prediction [111, 112, 113] to molecule generation [114, 115, 116]. To build this experiment, we select multiple-choice questions from our dataset and use one of the incorrect choices as context.

For example, given this question:

```
Which molecule listed here is most likely to have a rat microsomal
stability in mL/min/kg approximately equal to 1.26?
C1(C)=NN(C)C2=NC(C3C=CN=CC=3)=CC(=C12)C(=O)O
C12=NC(=CC(C(=O)O)=C2C(=NN1C)C)C(C)C
N1=CC=C(C2N=C3ON=C(C3=C(C(O)=O)C=2)CCC)C=C1
C1C(C2N=C3C(=C(C(O)=O)C=2)C(=NN3C2N=CC=CC=2)C)C1
```

We create an ICL equivalent of this task by taking one of the incorrect choices (highlighted in red) and using it as context in the question:

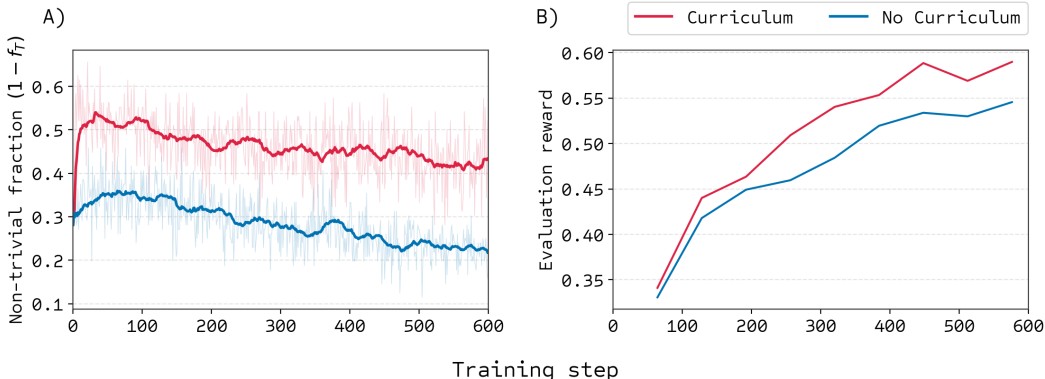

Figure S6: RL training dynamics of reaction prediction specialist models, one with and one without an online curriculum. A) the fraction of non-trivial groups seen during training (faint lines are raw data; solid are a 30-step moving average). B) the evaluation set reward, computed every 64 steps.

```
Considering C1(C)=NN(C)C2=NC(C3C=CN=CC=3)=CC(=C12)C(=O)O has a
measured rat microsomal stability in mL/min/kg of 1.03, which
candidate modification listed would most effectively increase this
property?
N1=CC=C(C2N=C3ON=C(C3=C(C(O)=O)C=2)CCC)C=C1
C1C(C2N=C3C(=C(C(O)=O)C=2)C(=NN3C2N=CC=CC=2)C)C1
C12=NC(=CC(C(=O)O)=C2C(=NN1C)C)C(C)C
```

To ensure that any observed performance improvement is not simply due to a reduced number of answer choices, we also modify the original question by removing the same incorrect option used as context in the ICL version. This way, both the standard and ICL queries present the same number of choices, preserving the same baseline probability of selecting the correct answer by chance (random baseline shown in Figure 3B.

### F.5 Human expert benchmarks

In Table S2, we report the breakdown of human expert performance on our test set.

| Task | Accuracy |
|------|----------|
| Molecular formula | $0.30^{0.60}_{0.13}$ |
| Functional group | $0.13^{0.30}_{0.00}$ |
| Reaction prediction | $0.41^{0.72}_{0.08}$ |
| Solubility edit | $0.05^{0.12}_{0.00}$ |
| Retrosynthesis | $0.00^{0.00}_{0.00}$ |
| Safety | $0.40^{0.52}_{0.28}$ |
| ADME | $0.32^{0.48}_{0.24}$ |
| LD50 | $0.43^{0.64}_{0.24}$ |
| pKa | $0.35^{0.48}_{0.20}$ |

Table S2: Human expert performance on the test set. Four contractors were tasked with solving each question, with no tools besides ChemDraw. We report the average accuracy, as well as the minimum and maximum scores, as sub- and super-scripts respectively.

```
Rubric provided for trace evaluations
```

**Genuineness:** The provided trace does not contain any contrived or performative reasoning.

**Faithful:** The model arrived at an answer based on the reasoning trace only, and did not make any sudden leaps of judgment.

**Exploration:** The trace displays examples of non-linear reasoning, self-reflection, or backtracking in its reasoning.

```
Pick the most suitable assessment for each metric:
1) Strongly disagree 2) Somewhat disagree 3) Neither agree
nor disagree 4) Somewhat agree 5) Strongly agree
```

Figure S7: Four expert evaluators were provided with this rubric to assess the "quality" of 15 traces from `ether0` and 15 traces from DeepSeek-R1.

| Non-Contrived / Genuine | S. Agree | Agree | Neutral | Disagree | S. Disagree |
|---|---|---|---|---|---|
| DeepSeek-r1 | 28% | 23% | 7% | 8% | 33% |
| ether0 | 45% | 23% | 12% | 15% | 5% |
| **Faithful** | **S. Agree** | **Agree** | **Neutral** | **Disagree** | **S. Disagree** |
| DeepSeek-r1 | 33% | 25% | 8% | 20% | 13% |
| ether0 | 50% | 23% | 10% | 13% | 3% |
| **Explorative** | **S. Agree** | **Agree** | **Neutral** | **Disagree** | **S. Disagree** |
| DeepSeek-r1 | 20% | 32% | 18% | 13% | 17% |
| ether0 | 13% | 28% | 35% | 5% | 18% |

Table S3: Expert evaluation of reasoning traces generated by `ether0` and DeepSeek-R1.

## F.6 Human evaluation

We conducted two sets of expert evaluations: 1) human baselines on a set of held-out open-ended and multiple-choice type questions, 2) `ether0` trace evaluations.

For the first set of evaluations (human baselines), we recruited four expert evaluators: two with PhDs in organic chemistry, one with a PhD in chemical engineering, and one PhD candidate in organic chemistry. Evaluators were instructed to respond using only the SMILES representation of the target molecule, without relying on external tools for assistance in answering. However, tools for visualizing SMILES as chemical structures were allowed. Tasks considered impossible to accomplish without the use of tools were flagged by the evaluators and excluded from the final analysis. Each evaluator was given 200 open-ended and/or multiple-choice questions from our held-out evaluation set, and was compensated \$10 per question completed. Their performance is compared with `ether0` and other frontier models in Figure 2.

For the second set of evaluations, we recruited another group of expert evaluators: three with PhDs in organic chemistry, and one with a PhD in chemical engineering. The evaluators were provided with a rubric to assess the reasoning traces generated by `ether0` and DeepSeek-R1 (see Figure S7). Each evaluator was given 30 reasoning traces from both model (15 from each). Compensation was \$10 per completed trace evaluation. Full comparison results are shown in Table S3. More experts disagreed that DeepSeek-r1 demonstrated non-contrived and faithful reasoning, although they noted it showed more extensive exploration. This is unsurprising, as DeepSeek-r1's reasoning traces were much longer than `ether0`'s.

