# OpenReview forum: "Training a Scientific Reasoning Model for Chemistry"
_NeurIPS.cc/2025/Conference — NeurIPS 2025 poster_

### Official Review · Reviewer_dvLR · 2025-06-30

**Clarity:** 4
**Significance:** 3
**Originality:** 3
**Rating:** 5
**Confidence:** 3

**Summary:**

This paper introduces Science0-c, a 24-billion parameter large language model (LLM) specifically trained to solve complex chemistry problems.  Based on the Mistral-Small-24B model, Science0-c is designed to reason in natural language and provide answers in chemical structure formats like SMILES. The training process involved a multi-stage approach combining supervised fine-tuning on chain-of-thought examples, training task-specific "specialist" models via RL, distilling their knowledge into a single "generalist" model, and then further training the generalist model with RL.  The authors report that Science0-c significantly outperforms existing general-purpose chemistry models, frontier LLMs like Claude and GPT models, and even human experts on a range of open-ended chemistry tasks.

**Questions:**

1. The authors' commitment to open-sourcing the code, dataset, and model is a significant contribution. However, the anonymous repository link provided for review appears to be empty. Given the highly complex, multi-stage training pipeline, reproducing these results presents a considerable challenge. For the benefit of the review process and future reproducibility, it would be immensely helpful if the authors could update the repository with the functional code. Furthermore, providing samples from the training data for each stage, along with a few examples of the final model's reasoning texts, would be invaluable for a thorough analysis.

2. The training curves in Figure 2a indicate that for several tasks, the task-specific reinforcement learning phase begins with a success rate close to zero. Standard RL is often inefficient in such a sparse reward setting, as the policy may struggle to generate successful samples to initiate the learning gradient. Could the authors elaborate on how the model was able to overcome this challenge?

3. In the data efficiency analysis, the paper compares Science0-c to the Molecular Transformer, a 2019 model. While this model was foundational, have the authors considered comparing their results against more contemporary state-of-the-art models for reaction prediction? A comparison against a more recent baseline would provide a stronger validation of the model's claimed superiority in data efficiency and performance.

4. The study primarily employs a binary reward signal (0 or 1) for task success. Could the authors discuss the rationale for this choice over a continuous reward function? What are the authors' thoughts on the challenges and opportunities in designing more sophisticated reward functions for complex scientific reasoning tasks in chemistry?

**Ethical Concerns:**

["NO or VERY MINOR ethics concerns only"]

**Final Justification:**

Most of my concerns are addressed by the authors. I think this is an important work for the chemical area. Due to several questions I proposed, such as "performance gap with specialized models" and "loss of general capabilities", are not addressed, I maintain my rating of 5.

**Limitations:**

1. Loss of General Capabilities: The intensive RL training focused on chemistry tasks has removed the general instruction-following and conversational abilities of the base Mistral model.
2. No Tool Use: The model was not trained to use external tools, which is a common and critical part of modern computational chemistry workflows.  Integrating tool-calling abilities is left for future work

**Quality:**

3

**Strengths And Weaknesses:**

strengths
1. Novel Application: The research successfully extends the application of "reasoning models," previously focused on math and programming, to the scientific domain of chemistry, demonstrating their potential beyond logic-based fields.
2. Superior Performance: Science0-c shows state-of-the-art performance, outperforming other large-scale models and human experts on several challenging, open-ended chemical reasoning tasks.  This is a significant achievement, particularly on complex problems like structure elucidation and retrosynthesis.
3. Methodological Rigor: The authors provide a detailed and transparent account of their multi-stage training pipeline, including supervised fine-tuning, specialist RL, distillation, and the use of an advantage-based curriculum.  The inclusion of ablation studies, such as comparing reasoning vs. non-reasoning models, strongly supports their central claim that explicit reasoning is beneficial.

weaknesses
1. Reasoning Quality Degradation: The study honestly reports that as RL training progresses and accuracy increases, the quality of the model's reasoning text can degrade, with the emergence of typos, non-English characters, and nonsensical formatting.
2. Performance Gap with Specialized Models: While the model demonstrates impressive performance, outperforming both proprietary and open-source chemistry LLMs, it still lags behind state-of-the-art specialized models. These specialized models, which are trained exclusively for a single purpose (such as retrosynthesis), often represent the true state-of-the-art in performance for that specific application. Therefore, for users requiring the absolute highest accuracy on a single, well-defined problem, this generalist reasoning model may not be the optimal choice.

---

> ### Author Rebuttal · Authors · 2025-07-29
>
> We thank the reviewer for the constructive review and helpful observations. We address the weaknesses and questions raised by the reviewer one by one below.
>
> **Reasoning Quality Degradation: The study honestly reports that as RL training progresses and accuracy increases, the quality of the model's reasoning text can degrade, with the emergence of typos, non-English characters, and nonsensical formatting.**
>
> This is indeed an interesting phenomenon that we observed. While the underlying reason is not entirely clear to us, we have reported it as a potentially interesting direction for future research.
>
> **Performance Gap with Specialized Models: While the model demonstrates impressive performance, outperforming both proprietary and open-source chemistry LLMs, it still lags behind state-of-the-art specialized models. These specialized models, which are trained exclusively for a single purpose (such as retrosynthesis), often represent the true state-of-the-art in performance for that specific application. Therefore, for users requiring the absolute highest accuracy on a single, well-defined problem, this generalist reasoning model may not be the optimal choice.**
>
> In our evaluation with specialized models, we focused on reaction prediction using the Molecular Transformer (MT) model. We acknowledge that other specialized models could be considered for other tasks, and we are open to noting this in the limitations. More specifically, that a performance gap may exist with respect to state-of-the-art models for certain specific tasks.
>
> **The authors' commitment to open-sourcing the code, dataset, and model is a significant contribution. However, the anonymous repository link provided for review appears to be empty. Given the highly complex, multi-stage training pipeline, reproducing these results presents a considerable challenge. For the benefit of the review process and future reproducibility, it would be immensely helpful if the authors could update the repository with the functional code. Furthermore, providing samples from the training data for each stage, along with a few examples of the final model's reasoning texts, would be invaluable for a thorough analysis.**
>
> We sincerely apologize for the oversight regarding the anonymous repository link. Thank you for pointing this out.
>
> We have now corrected the issue, the link is fully functional and the repository should allow for a review of our training codebase, including the scripts used for SFT and GRPO training, which are located in the *scripts* directory. Please note that we have removed all references to our org to comply with the double blind review, which may have caused degradation in the code's functionality.
>
> Finally, we have included in the anonymized github repo a representative sample of the final model’s reasoning outputs. These examples can be found inside the *data/molrqa_model_answers* directory.
>
> **The training curves in Figure 2a indicate that for several tasks, the task-specific reinforcement learning phase begins with a success rate close to zero. Standard RL is often inefficient in such a sparse reward setting, as the policy may struggle to generate successful samples to initiate the learning gradient. Could the authors elaborate on how the model was able to overcome this challenge?**
>
> While it is true that RL algorithms can be inefficient in sparse reward settings, RL training remains viable as long as some non-zero reward signal is eventually encountered. In practice, RL can learn effectively even from rare success signals, it just tends to start slower, as evidenced in Figure 2a. Despite initially low success start rate, we found that the model is able to bootstrap off rare successful trajectories, and this was often sufficient to get learning off the ground.
>
> Moreover, we incorporate a curriculum learning strategy based on advantage-weighted sampling. Specifically, during training we preferentially sample prompts where the estimated advantage is non-zero. This naturally biases learning toward trajectories that are already yielding some signal, thus helping the model focus on parts of the prompt distribution where learning progress is possible. This targeted sampling strategy mitigates the exploration problem and accelerates convergence in sparse reward regimes.
>
> **In the data efficiency analysis, the paper compares Science0-c to the Molecular Transformer, a 2019 model. While this model was foundational, have the authors considered comparing their results against more contemporary state-of-the-art models for reaction prediction? A comparison against a more recent baseline would provide a stronger validation of the model's claimed superiority in data efficiency and performance.**
>
> Despite being published in 2019, Molecular Transformers (MT) is still among the best models for reaction prediction, reaching 88.6 %  pass@1 accuracy  on USPTO_MIT [1]. Additionally, MT is a relatively small model, making it easier to perform multiple studies with it. However, to illustrate our performance against more up-to-date models in diverse chemistry tasks, we also compared Science0-c to ChemDFM (2024).
>
> **The study primarily employs a binary reward signal (0 or 1) for task success. Could the authors discuss the rationale for this choice over a continuous reward function? What are the authors' thoughts on the challenges and opportunities in designing more sophisticated reward functions for complex scientific reasoning tasks in chemistry?**
>
> We experimented with continuous reward functions on several tasks, including reaction prediction and retrosynthesis, but these attempts did not improve over our binary baseline in most cases. Designing effective continuous rewards is complex, often considered almost an art, and requires deep domain expertise and significant time investment. Even then, it may fail to provide useful learning signals. Therefore, we chose not to pursue this direction further after a certain number of unsuccessful attempts.
>
> Given these challenges and the limited gains from our early trials, we adopted binary rewards across all tasks except one. The exception is the specialist trained on molecular formula, functional group, and elucidation tasks, where the reward was set to 0.5 if the formula was correct and 1.0 if all constraints were met. We found that this setting improved performance over the binary baseline.
>
> **Loss of General Capabilities: The intensive RL training focused on chemistry tasks has removed the general instruction-following and conversational abilities of the base Mistral model.**
>
> This is true. In particular, the fact of focusing on tasks whose expected answer is always a SMILES, has removed the conversational abilities of the base mistral model.
>
> **No Tool Use: The model was not trained to use external tools, which is a common and critical part of modern computational chemistry workflows. Integrating tool-calling abilities is left for future work**
>
> This is also correct, tool use was left for future work. We believe integrating tool-calling has strong potential to unlock even more accurate and useful capabilities, especially for real-world chemistry workflows, but in this work we focused on verifying whether reasoning alone was helpful, and left the addition of tools for future work.
>
> **References:**
>
> [1] Jin, Wengong, et al. "Predicting organic reaction outcomes with weisfeiler-lehman network." Advances in neural information processing systems 30 (2017).

---

> > ### Comment · Reviewer_dvLR · 2025-08-04
> >
> > Thanks for your response. Since the limitations I propose, such as Performance Gap with Specialized Models, still exist, I will maintain my rating.

---

### Official Review · Reviewer_ZRWC · 2025-07-01

**Clarity:** 4
**Significance:** 3
**Originality:** 3
**Rating:** 5
**Confidence:** 4

**Summary:**

This paper presents Science0-c, a 24B parameter reasoning LLM fine-tuned for chemistry using reinforcement learning on 577,790 verifiable tasks involving molecule design, modification, completion, and synthesis. The model demonstrates strong performance across open-ended and multiple-choice tasks, outperforming general-purpose LLMs, humans, and prior chemistry-specific models.

**Questions:**

see weakness

**Ethical Concerns:**

["NO or VERY MINOR ethics concerns only"]

**Final Justification:**

The authors have provided a more thorough qualitative analysis of the model’s reasoning traces and added a discussion of Science0-c's potential role in drug discovery workflows. Although the model does not transfer well to tasks outside its training distribution, it performs well on the proposed chemistry tasks and shows thoughtful design for those use cases.

**Limitations:**

yes

**Quality:**

4

**Strengths And Weaknesses:**

Strength:

-The paper tackles the underexplored problem of scientific reasoning in chemistry using LLMs

-Strong empirical performance on diverse and challenging chemistry tasks

-Science0-c outperforms both general-purpose LLMs (e.g., GPT-o1, Claude) and specialized chemistry models (e.g., TxGemma, ChemDFM)

-Detailed training pipeline with reinforcement learning, task specialization, and generalist distillation.

-Open-sourcing of model, code, and reward functions


Weakness:

-It would be helpful if the authors briefly discuss the broader utility of Science0-c, such as its potential use in drug discovery pipelines

-The reasoning traces provide useful insight into the model’s thought process, making Science0-c partially interpretable by design. However, a small-scale case study—such as having domain experts annotate a sample of reasoning traces for correctness, relevance, or use of chemical knowledge—could help strengthen the scientific trustworthiness.

-After being trained on this set of chemistry tasks, can the model generalize to unseen tasks by leveraging the chemical knowledge it has learned?

---

> ### Author Rebuttal · Authors · 2025-07-29
>
> We thank the reviewer for the constructive review and helpful remarks. We address the weaknesses raised by the reviewer one by one below.
>
> **It would be helpful if the authors briefly discuss the broader utility of Science0-c, such as its potential use in drug discovery pipelines.**
>
> We agree that highlighting the broader utility of Science0-c in the context of drug discovery is important to illustrate its relevance in practical settings and the potential impact beyond the reported benchmarks. We will incorporate this perspective into the introduction of our manuscript.
>
> Specifically, Science0-c supports key stages of the drug discovery pipeline: it can generate candidates during hit discovery, it enables molecule editing in hit-to-lead to improve potency, selectivity, or physicochemical properties, and it contributes to lead optimization by refining compounds to enhance efficacy, reduce toxicity, and improve ADMET profiles. All while being aware of synthesizability.
>
> Additionally, and unlike models like molt5 or chemformer, Science0-c works directly in natural language, which facilitates integration into drug discovery workflows through intuitive interaction, and its reasoning traces can help medicinal chemists understand the justification of proposed molecules, especially in high-stakes decisions.
>
> **The reasoning traces provide useful insight into the model’s thought process, making Science0-c partially interpretable by design. However, a small-scale case study—such as having domain experts annotate a sample of reasoning traces for correctness, relevance, or use of chemical knowledge—could help strengthen the scientific trustworthiness.**
>
> We actually conducted a qualitative analysis of the reasoning traces between DeepSeek-r1 and Science0-c. While this comparative analysis was not included in the original manuscript (only the results for Science0-c were reported in the supplementary information section 3), upon further review we recognize the value of reporting the results also for DeepSeek-r1 and will add them as well.
>
> As explained in the supplementary information, we asked human experts to evaluate how much they agreed that the traces from both models were non-contrived/genuine (in the sense that they did not exhibit performative thinking without clear justification given the problem under consideration), faithful (i.e., the answer is consistent with the reasoning trace), and whether the model performed an extensive exploration before answering. Experts had five response options: strongly agree, somewhat agree, neither agree nor disagree, somewhat disagree, and strongly disagree.
>
> The results showed that more experts disagreed with the claim that DeepSeek-r1 demonstrated non-contrived and faithful reasoning, although they noted it did have more extensive exploration. This is somewhat expected, given the average length of DeepSeek-r1’s reasoning traces being much longer than Science0-c traces.
>
> We report our results here and will include them in the final version of the manuscript:
>
> Non-Contrived/Genuine:
> | Model            | S. Agree | Agree  | Neutral      | Disagree | S Disagree
> |-------------------|-------------|----------|-----------------|-------------|-----------------|
> | DeepSeek-r1 | 28%       | 23%    | 7%              | 8%        | 33%             |
> | Science0-c    | 45%       | 23%    | 12%             | 15%       | 5%              |
>
> Faithful:
> | Model            | S. Agree | Agree  | Neutral      | Disagree | S Disagree
> |-------------------|-------------|----------|-----------------|-------------|-----------------|
> | DeepSeek-r1 | 33%       | 25%   | 8%               | 20%       | 13%            |
> | Science0-c    | 50%       |23%     | 10%             | 13%        | 3%             |
>
> Explorative:
> | Model            | S. Agree | Agree  | Neutral      | Disagree | S Disagree
> |-------------------|-------------|----------|-----------------|-------------|-----------------|
> | DeepSeek-r1 | 20%      | 32%    | 18%             | 13%       | 17%             |
> | Science0-c    | 13%       | 28%    | 35%             | 5%         | 18%            |
>
> **After being trained on this set of chemistry tasks, can the model generalize to unseen tasks by leveraging the chemical knowledge it has learned?**
>
> First, we would like to emphasize that our goal was to design a dataset of specific, practical, and challenging problems that are useful in real research. Our objective was not to train a general-purpose model capable of generalizing to new tasks, which is why our evaluation focuses on a held-out subset of problems from the same dataset.
>
> That being said, we did evaluate the model on some tasks outside the training task distribution, such as ChemBench [1] and chebi-20 [2]. We confirmed that the performance of Science0-c on these unseen tasks is limited. Science0-c obtains 44% on ChemBench and 31% on chebi-20 pass@1, in both cases far from the state-of-the-art. Nonetheless, nothing suggests that these additional tasks could not be incorporated into the dataset and learned using the approach described in our work, potentially leveraging transfer learning from the other trained tasks. We acknowledge that this information is important and will make these limitations clearer by mentioning these results.
>
> We also wanted to ensure that the model was robust to variations in phrasing. This is relevant to generalise within the trained tasks. To evaluate this, we ran an experiment where we asked a frontier model (Gemini) to rephrase the test questions and found no significant performance differences compared to the original versions. Similarly, we added distracting information to the prompts and again observed no meaningful performance change. We also plan to include it in a final version of the paper.
>
> **References:**
>
> [1] Mirza, Adrian, et al. "Are large language models superhuman chemists?." arXiv preprint arXiv:2404.01475 (2024).
>
> [2] Edwards, Carl, ChengXiang Zhai, and Heng Ji. "Text2mol: Cross-modal molecule retrieval with natural language queries." Proceedings of the 2021 Conference on Empirical Methods in Natural Language Processing. 2021.

---

> > ### Comment · Reviewer_ZRWC · 2025-08-05
> >
> > Thank you for your response. I no longer have concerns and maintain a positive attitude toward this paper.

---

### Official Review · Reviewer_ha5g · 2025-07-01

**Clarity:** 4
**Significance:** 3
**Originality:** 2
**Rating:** 5
**Confidence:** 4

**Summary:**

This paper introduces Science0-c, an LLM specifically developed for scientific reasoning in chemistry. The authors constructed a large dataset of ~570K chemistry problems covering 17 tasks such as reaction prediction, SMILES completion, and molecule design under specific constraints. Science0-c is trained from the Mistral-Small-24B-Instruct-2501 base model using this dataset with a multi-stage training procedure that interleaves distillation and GRPO.

Experimental evaluation on Science0-c's test set shows that the proposed model outperforms both frontier and domain-specific models.

**Questions:**

* What're the exact sizes of datasets used during various phases of the training pipeline? What percentage of the data is filtered out due to quality concerns at each stage?

* Is the second distillation performed on the warm started model or directly on the base mistral model?

* Please explain the subtasks within MCQ tasks like Safety, Scent, Receptor Binding, ADME, and pKa.

* Is there a reason the 'Human' baseline was not included in the MCQ results? Please include it if possible.

* Are there plans to release the full code and datasets?

**Ethical Concerns:**

["NO or VERY MINOR ethics concerns only"]

**Final Justification:**

This work on reasoning for chemistry tasks is quite promising. The authors offer a thorough overview of their data, training approach, and evaluation. The rebuttal reasonably addressed my earlier concerns about novelty and performance on out-of-distribution tasks.

**Quality:**

3

**Strengths And Weaknesses:**

**Strengths:**

* The authors carefully created a large dataset of chemistry problems (MCQ & open-ended) that covers various chemistry tasks from data sources like ChEMBL, COCONUT and Pubchem. The initial dataset and distillation dataset would be valuable for future research if publicly released.

* The detailed description of the training methodology, including SFT, task-specific RL, distillation, and a final all-task RL phase, demonstrates a well thought-out and effective approach for developing specialized reasoning models.

* Including "reasonable molecule" check in GRPO verifier is an interesting strategy to ensure the usefulness and plausability of the molecules generated by Science0-c

* The paper's detailed analysis on cognitive behaviors like verification and backtracking in reasoning traces during training, and their correlation with task performance, provides valuable insights into the internal workings and effectiveness of reasoning models.

* Overall, it's a well written and easy to read paper


**Weaknesses:**

* Although the experimental results show the significant benefits of a reasoning model for chemistry, the training approach relies on established techniques like SFT, GRPO, and distillation, and lacks novelty.

* Science0-c is mainly benchmarked on its own test set. Additional evaluation on existing chemistry benchmarks like ChemLLMBench could be useful for a more comprehensive assessment.

* Ablation studies on vanilla SFT over the full ~570K dataset, and SFT + all-task RL without the task-specific RL phase, would further help to understand the specific benefits and contributions of the proposed multi-stage training procedure.

---

> ### Author Rebuttal · Authors · 2025-07-29
>
> We thank the reviewer for the constructive review and helpful remarks. We address the weaknesses raised by the reviewer one by one below.
>
> **Although the experimental results show the significant benefits of a reasoning model for chemistry, the training approach relies on established techniques like SFT, GRPO, and distillation, and lacks novelty.**
>
> While it is true that our training pipeline builds on established techniques like SFT and GRPO, it also incorporates several novel components that distinguish it from the initial wave of reasoning model research released in early 2025. These include advantage-based curriculum training (with some concurrent work using similar yet different approaches to tackle the same problem, like Llama-Nemotron [1] and DAPO [2]), rephrasing demonstration trajectories using our target model to better align the token distribution, interleaving multiple rounds of SFT and GRPO, and the use of specialist models (a similar idea was later incorporated into a revised version of the DeepSeek-V3 [3] paper).
>
> **Science0-c is mainly benchmarked on its own test set. Additional evaluation on existing chemistry benchmarks like ChemLLMBench could be useful for a more comprehensive assessment.**
>
> First, we would like to emphasize that one of our goals was to design a dataset of specific, practical, and challenging problems that are useful in real research. Our objective was not to train a general-purpose model capable of generalizing to new tasks, which is why our evaluation focuses on a held-out subset of problems from the same dataset.
>
> However, we did evaluate the model on some tasks outside the training task distribution, such as ChemBench [4] and the suggested chebi-20 [5]. We confirmed that the performance of Science0-c on these unseen tasks is limited. Science0-c obtains 44% on ChemBench and 31% on chebi-20 pass@1, in both cases far from the state-of-the-art. Nonetheless, nothing suggests that these additional tasks could not be incorporated into the dataset and learned using the approach described in our work, potentially leveraging transfer learning from the other trained tasks. We acknowledge that this information is important and will make these limitations clearer by mentioning these results.
>
> We also wanted to ensure that the model was robust to variations in phrasing. This is relevant to generalise within the trained tasks. To evaluate this, we ran an experiment where we asked a frontier model (Gemini) to rephrase the test questions and found no significant performance differences compared to the original versions. Similarly, we added distracting information to the prompts and again observed no meaningful performance change. We also plan to include it in a final version of the paper.
>
> **Ablation studies on vanilla SFT over the full ~570K dataset, and SFT + all-task RL without the task-specific RL phase, would further help to understand the specific benefits and contributions of the proposed multi-stage training procedure.**
>
> We did not perform SFT over the full 577,790 dataset because reasoning traces are not available for all examples. Initially only those solved by DeepSeek-r1 (14,021 samples) and later those solved by the specialists (186,010 samples) are available. The total number of solved problems with reasoning traces available is 200,031.
>
> We did perform training on the 200,031 examples available (a significant subset of the full dataset), which corresponds to the second distillation phase of our pipeline. However, we observed that we could further push performance of the model with the final RL phase.
>
> Training on just (q, a) pairs (without reasoning traces) corresponds to our provided non-reasoning SFT baseline.
>
> As for SFT followed by all-task RL without task-specific RL, we did not scale this configuration fully due to early signs of poor learning dynamics. In particular, some task categories such as molecular formula and functional group showed little to no learning under this setup. We acknowledge that this information is relevant and will include it in the manuscript for completeness.
>
> **What're the exact sizes of datasets used during various phases of the training pipeline? What percentage of the data is filtered out due to quality concerns at each stage?**
>
> The complete train dataset contains 577,790 samples, as detailed in **Table 1**.
>
> 1. Initial Supervised Fine-Tuning (SFT):
>    - Conducted on 14,021 samples originally solved by DeepSeek-r1.
> 2. Specialist Training with GRPO:
>    - GRPO training used the entire train dataset, split by task.
> 3. Filtering specialist data
>    - Specialists collectively solved 3,830,728 questions (with repeated questions).
>    - From these questions, we retained only the most recent solution per unique question, resulting in 201,957 samples.
>    - We found that ~13% of the samples were filtered out due to quality concerns but in almost all cases we could find an earlier instance in which the model solved the question with high quality, and used that one instead. To enhance clarity, we will add this information to the paper. The final dataset after the quality filtering was 200,031 samples.
>    - The specialist solved questions included the 14,021 questions originally solved by DeepSeek-r1, so the specialist models effectively contributed to solve 186,010 new high-quality samples.
> 4. Second SFT Phase:
>    - SFT was performed on the 200,031 samples.
> 5. Final RL Phase:
>    - GRPO on the full dataset of 577,790 samples again.
>
> **Is the second distillation performed on the warm started model or directly on the base mistral model?**
>
> The second distillation was performed on the base mistral model. We will be more clear about that, thank you.
>
> **Please explain the subtasks within MCQ tasks like Safety, Scent, Receptor Binding, ADME, and pKa.**
>
> As suggested by the reviewer, we will add formal definitions for each subtask, likely in the Supplementary Information (SI) due to space limitations in the main text.
>
> *Safety:* Select the molecule whose structure most strongly aligns with or deviates from a specified safety-related property, such as toxicity, flammability, or hazard classification.
> *Scent:* Identify the molecule most likely to exhibit a specific olfactory attribute (e.g., meaty, spicy, oily).
> *Blood-brain barrier:* Determine which molecule is or isn’t structurally likely to penetrate the blood-brain barrier based on reference behavior.
> *Receptor Binding:* Identify the molecule whose structure most likely lacks binding affinity or activity for a specified biological receptor target.
> *ADME:* Choose the molecule expected to improve or match a specified absorption, distribution, metabolism, or excretion property based on structural modifications.
> *Aqueous Solubility:* Select the molecule whose structure leads to a targeted increase or decrease in water solubility.
> *LD50:* Select the molecule whose structure corresponds to a specified rat oral LD50 value, reflecting its relative acute toxicity level.
> *pKa:* Select the molecule whose structure aligns with a specified increase, decrease, or target value of pKaH1.
> *Photoswitches:* Select the molecule whose E isomer exhibits a target π–π* transition wavelength.
>
> **Is there a reason the 'Human' baseline was not included in the MCQ results? Please include it if possible.**
>
> We focused our human evaluations on the open-ended questions due to resource constraints. If the reviewer sees value in including human performance on MCQs, we're open to running them and incorporating the results in the final version.
>
> **Are there plans to release the full code and datasets?**
>
> The anonymized link in the SI includes our scripts for SFT and GRPO training, located in the *scripts* directory. All references to our organization have been removed to comply with double-blind review. We also have plans to release the code for our reward functions, prompt templates, data utilities, and the dataset.
>
> **References:**
>
> [1] Bercovich, Akhiad, et al. "Llama-nemotron: Efficient reasoning models." arXiv preprint arXiv:2505.00949 (2025).
>
> [2] Yu, Qiying, et al. "Dapo: An open-source llm reinforcement learning system at scale." arXiv preprint arXiv:2503.14476 (2025).
>
> [3] Liu, Aixin, et al. "Deepseek-v3 technical report." arXiv preprint arXiv:2412.19437 (2024).
>
> [4] Mirza, Adrian, et al. "Are large language models superhuman chemists?." arXiv preprint arXiv:2404.01475 (2024).
>
> [5] Edwards, Carl, ChengXiang Zhai, and Heng Ji. "Text2mol: Cross-modal molecule retrieval with natural language queries." Proceedings of the 2021 Conference on Empirical Methods in Natural Language Processing. 2021.

---

> > ### Comment · Reviewer_ha5g · 2025-08-05
> >
> > Thanks for the detailed response. Regarding the novelty in the training approach, I agree the paper introduces a few things like the advanced-based curriculum. I’ll update my score to reflect that.
> >
> > It would be good to add human/expert baselines to the MCQ results. I expect that's easier to do than most of the open-ended questions.

---

> > > ### Author Response · Authors · 2025-08-06
> > >
> > > We would like to let the reviewer know that we are committed to including human expert baseline evaluations for the MCQ tasks. We agree will provide additional context for interpreting our model's performance.

---

### Official Review · Reviewer_hCQw · 2025-07-02

**Clarity:** 3
**Significance:** 3
**Originality:** 3
**Rating:** 5
**Confidence:** 3

**Summary:**

This paper introduces Science0-c, a 24-billion parameter LLM designed for chemical reasoning. It's built upon the Mistral-Small-24B model and has been specifically fine-tuned to understand natural language prompts and respond with chemical structures in SMILES format.

**Questions:**

Regarding Figure 2, the legend does not define the blue cross symbol. Though I can guess the author skip tasks related to SMILES for human, it should be explicitly stated in the caption for clarity.

**Ethical Concerns:**

["NO or VERY MINOR ethics concerns only"]

**Final Justification:**

My primary reservations were initially centered on two aspects: the validity of the retrosynthesis dataset and the statistical significance of the human baseline. The authors clarified that reaction templates were used to ensure the training data was grounded in experimentally feasible chemistry. They explained that the Molecular Transformer was a necessary tool for validating the novel reactions proposed by the model, a task for which experimental data does not yet exist. Crucially, they have committed to explicitly discussing the known limitations of this validation approach in the final manuscript. This transparency is a responsible and satisfactory resolution. Also, the authors have agreed to add error bars to the human evaluation plots to report performance variability. This directly addresses my concern and provides the necessary context to fairly interpret the comparison between the model and human experts.

**Limitations:**

Yes.

**Quality:**

3

**Strengths And Weaknesses:**

Strength:
1. A key strength is the demonstration that a general reasoning model can be effectively specialized for chemistry, which is significantly more data-efficient than creating a specialized model from scratch.
2. The discussion section is particularly robust, offering a well-rounded analysis of both the model's groundbreaking capabilities and its current constraints.

Weakness:
1. The validity of the retrosynthesis dataset is a point of concern. The authors mention the use of Molecular Transformer in the main text and reaction templates in the SI, rather than using experimentally verified reactions. This approach introduces a potential gap between the computational predictions and practical laboratory outcomes.
2. The human baseline evaluation was conducted with a small panel of four to five experts, which limits the statistical significance of these results. This small sample size weakens claims about the model's performance relative to human experts. To strengthen the comparison, the authors should, at a minimum, report the variability in human performance by including error bars.

---

> ### Author Rebuttal · Authors · 2025-07-29
>
> We thank the reviewer for the constructive review and helpful observations. We address the weaknesses and questions raised by the reviewer one by one below.
>
> **The validity of the retrosynthesis dataset is a point of concern. The authors mention the use of Molecular Transformer in the main text and reaction templates in the SI, rather than using experimentally verified reactions. This approach introduces a potential gap between the computational predictions and practical laboratory outcomes.**
>
> We used reaction templates to generate the dataset to ensure that the training data consists of experimentally synthesizable molecules. However, since Science0-c proposes new, previously unseen reactions, we required a more general method to validate its predicted reactions. To make the reward function broadly applicable, we used Molecular Transformers (MT) for reaction validation. We acknowledge that this approach has limitations, as MT itself is not without shortcomings. These limitations will be discussed more explicitly in the limitations section.
>
> **The human baseline evaluation was conducted with a small panel of four to five experts, which limits the statistical significance of these results. This small sample size weakens claims about the model's performance relative to human experts. To strengthen the comparison, the authors should, at a minimum, report the variability in human performance by including error bars.**
>
> We will include error bars in the final version of the plot to reflect variability in human performance. Meanwhile, we report here the standard deviation across human annotators for each task in the table below:
>
> | Task                                                               | std                   |
> |------------------------------------------------------------|---------------------|
> | Functional group                                            | 0.12                 |
> | Solubility edit                                                  | 0.06                 |
> | Reaction prediction                                        | 0.26                 |
> | Retrosynthesis                                               | 0.05                 |
> | Molecular formula                                          | 0.22                 |
>
> **Regarding Figure 2, the legend does not define the blue cross symbol. Though I can guess the author skip tasks related to SMILES for human, it should be explicitly stated in the caption for clarity.**
>
> This is correct, thanks for the observation. The blue cross indicates that human evaluations were not possible for the task, mostly because humans were not able to do things like SMILES completion. We'll add this clarification to the caption.

---

> > ### Comment · Reviewer_hCQw · 2025-08-02
> >
> > Thank you for your response; I believe this is very important work for specific chemical applications and will raise my score accordingly.

---

### Official Review · Reviewer_uSST · 2025-07-22

**Clarity:** 3
**Significance:** 2
**Originality:** 2
**Rating:** 4
**Confidence:** 4

**Summary:**

This work introduces a dataset of 577,790 experimentally-grounded chemistry problems, categorized into 17 distinct tasks. Each problem includes a solution that can be verified using KDESol, RDKit, or string matching. The authors also trained a 24-billion-parameter reasoning language model on this dataset, named Science0-c. Science0-c outperforms both general-purpose reasoning language models and specialized non-LLM models on chemistry-related tasks.

**Questions:**

Please refer to the Weaknesses above

**Ethical Concerns:**

["NO or VERY MINOR ethics concerns only"]

**Final Justification:**

Overall, the rebuttal addresses most of the issues and concerns raised in the discussion, so I have raised my score

**Limitations:**

Please refer to the Weaknesses above

**Quality:**

2

**Strengths And Weaknesses:**

**Strengths:**

- The paper addresses an important problem in the intersection of chemistry and reasoning language models.
- The presentation is clear and easy to follow.


**Weaknesses:**

- The evaluation is limited to a single benchmark consisting of holdout problems from the dataset. There is no assessment on tasks outside the training data, such as molecule captioning (e.g., ChEBI-20 [1]) or text-based molecule design (e.g., ChemLLMBench [2]).
- The contribution in terms of novelty is limited. The dataset is constructed from existing sources and verified using standard toolkits, and the training pipeline closely follows prior work on reasoning LLMs with minimal modification [3].
- The paper lacks a qualitative analysis of the reasoning behavior of Science0-c compared to general-purpose reasoning LMs. For example, what are the typical failure modes of general-purpose models, and how does Science0-c address or overcome them?

[1] Edwards, C., Zhai, C. and Ji, H., 2021. Text2Mol: Cross-modal molecule retrieval with natural language queries. *Proceedings of the 2021 Conference on Empirical Methods in Natural Language Processing*, pp.595–607.

[2] Guo, T., Guo, K., Nan, B., Liang, Z., Guo, Z., Chawla, N.V., Wiest, O. and Zhang, X., 2023. What can large language models do in chemistry? A comprehensive benchmark on eight tasks. In: *Proceedings of the 37th International Conference on Neural Information Processing Systems (NeurIPS 2023)*

[3] DeepSeek-AI, Guo, D., Yang, D., Zhang, H., Song, J., Zhang, R., Xu, R., Zhu, Q., Ma, S., Wang, P. *et al.*, 2025. *DeepSeek-R1: Incentivizing reasoning capability in LLMs via reinforcement learning*. arXiv preprint arXiv:2501.12948.

---

> ### Author Rebuttal · Authors · 2025-07-29
>
> Many thanks to the reviewer for the valuable feedback provided. Here, we would like to address the main concerns raised on the review:
>
> **The evaluation is limited to a single benchmark consisting of holdout problems from the dataset:**
>
> First, we would like to emphasize that one of our goals was to design a dataset of specific, practical, and challenging problems that are useful in real research. Our objective was not to train a general-purpose model capable of generalizing to new tasks, which is why our evaluation focuses on a held-out subset of problems from the same dataset.
>
> However, we did evaluate the model on some tasks outside the training task distribution, such as ChemBench [1] and the suggested chebi-20. We confirmed that the performance of Science0-c on these unseen tasks is limited. Science0-c obtains 44% on ChemBench and 31% on chebi-20 pass@1, in both cases far from the state-of-the-art. Nonetheless, nothing suggests that these additional tasks could not be incorporated into the dataset and learned using the approach described in our work, potentially leveraging transfer learning from the other trained tasks. We acknowledge that this information is important and will make these limitations clearer by mentioning these results.
>
> We also wanted to ensure that the model was robust to variations in phrasing. This is relevant to generalise within the trained tasks. To evaluate this, we ran an experiment where we asked a frontier model (Gemini) to rephrase the test questions and found no significant performance differences compared to the original wording. Similarly, we added distracting information to the prompts and again observed no meaningful performance change. We also plan to include it in a final version of the paper.
>
> Additionally we want to emphasize that we ensured a fair comparison with all baselines on the test set. We adapted each test set problem to match the input format and expectations of the open-source models. For example, for ChemDFM, we adjusted the prompt wording and applied SMILES canonicalization. Similar adaptations were made for TxGemma. Naturally, all models used their own system prompts. We are aware that frontier LLMs likely haven't been optimized for the specific tasks considered in our work. However, they have been trained to be robust to different phrasings, so we consider it reasonable to evaluate them with the same questions as our model, as no specific phrasing will lead to improved performance a priori.
>
> **The contribution in terms of novelty is limited.**
>
> Our work introduces a set of advances that enable the effective application of reasoning models to scientific tasks, while also offering broader insights into the training of general-purpose reasoning systems.
>
> We begin by hypothesizing that open-ended scientific problems benefit from reasoning-based models due to their nature grounded in structured reasoning. We validate this hypothesis by demonstrating strong performance on chemistry tasks, confirming that such models are well-suited for complex scientific reasoning. To the best of our knowledge, this had not been demonstrated before. Additionally, we show that a general model, which has potentially seen limited chemistry data during pre-training, can be effectively specialized for chemistry, which we believe to be not obvious.
>
> While it is true that our training pipeline builds on established techniques like SFT and GRPO, it also incorporates several novel components that distinguish it from the initial wave of reasoning model research released in early 2025. These include advantage-based curriculum training (with some concurrent work using similar yet different approaches to tackle the same problem, like Llama-Nemotron [2] and DAPO [3]), rephrasing demonstration trajectories using the base model intended for RL training (Mistral-Small-24B))  to better align the demonstration token distributions, interleaving multiple rounds of SFT and GRPO, and the use of specialist models (a similar idea was later incorporated into a revised version of the DeepSeek-V3 [4] paper).
>
> Although our data sources are publicly available, our work combines them with well-designed reward functions and a robust training strategy that allows achieving top performance on several difficult chemistry benchmarks. The design of this pipeline contributes to the progress of AI for Chemistry. Additionally, toolkit-based rewards often fail when used out-of-the-box, and avoiding reward hacking is especially hard in this setting. Below are two examples and how we addressed them:
>
> For example, in designing the retrosynthesis reward function, Science0-c first learned to exploit the rule that only one of the reaction components needed to be purchasable. It began suggesting reactions where it swapped a bromine atom with another halide (like chlorine or fluoride) on the target molecule. These transformations often used precursors that weren’t purchasable, but the presence of a common purchasable halide (e.g., chloride) allowed Science0-c to game the reward. This simple, high-reward hack inflated its performance without truly solving the retrosynthesis task. We resolved this by expanding the bloom filter to include a comprehensive list of laboratory reagents and enforcing stricter checks on all reactants’ purchasability and molecular transformations.
>
> As another example, when asked to generate molecules with specific atomic compositions (like C₃H₅O₂), Science0-c would often append unstable or synthetically infeasible groups, such as long peroxide or hydrazine tails, to meet atom count requirements. These additions fulfilled the letter of the task but produced unrealistic, explosive molecules. To combat this, we created a bloom filter of all three-atom-radius subgraphs found in known synthesized molecules and used this as a "reasonableness" check in the reward function. We further filtered out chemically infeasible ring structures and tuned the reward to discourage nitro groups and other problematic patterns, shifting from brittle heuristics to preference bonus rewards.
>
> Finally, our work also provides a thorough analysis of the trained model’s behavior, including data efficiency, failure modes, and reasoning patterns. These insights shed light on how reasoning models behave in scientific domains, offering guidance for future research.
>
> **The paper lacks a qualitative analysis of the reasoning behavior of Science0-c compared to general-purpose reasoning LMs.**
>
> Of all the models evaluated, only DeepSeek-r1 provides traces we can examine. Closed-source models do not disclose their exact reasoning patterns, while specialized models do not perform reasoning at all.
>
> We actually conducted a qualitative analysis of the reasoning traces between DeepSeek-r1 and Science0-c. While this comparative analysis was not included in the original manuscript (only the results for Science0-c were reported in the supplementary information section 3), upon further review we recognize the value of reporting the results also for DeepSeek-r1 as suggested by the reviewer and will add the full comparison.
>
> As explained in the supplementary information, we asked human experts to evaluate how much they agreed that the traces from both models were non-contrived/genuine (in the sense that they did not exhibit performative thinking without clear justification given the problem under consideration), faithful (i.e., the answer is consistent with the reasoning trace), and whether the model performed an extensive exploration before answering. Experts had five response options: strongly agree, somewhat agree, neither agree nor disagree, somewhat disagree, and strongly disagree.
>
> The results showed that more experts disagreed with the claim that DeepSeek-r1 demonstrated non-contrived and faithful reasoning, although they noted it did have more extensive exploration. This is somewhat expected, given the average length of DeepSeek-r1’s reasoning traces being much longer than Science0-c traces.
>
> We report our results here and will include them in the final version of the manuscript:
>
> Non-Contrived/Genuine:
> | Model            | S. Agree | Agree  | Neutral      | Disagree | S Disagree
> |-------------------|-------------|----------|-----------------|-------------|-----------------|
> | DeepSeek-r1 | 28%       | 23%    | 7%              | 8%        | 33%             |
> | Science0-c    | 45%       | 23%    | 12%             | 15%       | 5%              |
>
> Faithful:
> | Model            | S. Agree | Agree  | Neutral      | Disagree | S Disagree
> |-------------------|-------------|----------|-----------------|-------------|-----------------|
> | DeepSeek-r1 | 33%       | 25%   | 8%               | 20%       | 13%            |
> | Science0-c    | 50%       |23%     | 10%             | 13%        | 3%             |
>
> Explorative:
> | Model            | S. Agree | Agree  | Neutral      | Disagree | S Disagree
> |-------------------|-------------|----------|-----------------|-------------|-----------------|
> | DeepSeek-r1 | 20%      | 32%    | 18%             | 13%       | 17%             |
> | Science0-c    | 13%       | 28%    | 35%             | 5%         | 18%            |
>
> **References:**
>
> [1] Mirza, Adrian, et al. "Are large language models superhuman chemists?." arXiv preprint arXiv:2404.01475 (2024).
>
> [2] Bercovich, Akhiad, et al. "Llama-nemotron: Efficient reasoning models." arXiv preprint arXiv:2505.00949 (2025).
>
> [3] Yu, Qiying, et al. "Dapo: An open-source llm reinforcement learning system at scale." arXiv preprint arXiv:2503.14476 (2025).
>
> [4] Liu, Aixin, et al. "Deepseek-v3 technical report." arXiv preprint arXiv:2412.19437 (2024).

---

> > ### Author Response · Authors · 2025-08-05
> >
> > We appreciate the reviewer's thorough read of our paper. We have posted a detailed rebuttal above, and as the discussion period is drawing to a close, would like to ask for feedback from the reviewer. Have we addressed your concerns, and are there further studies you would like to see?

---

> > > ### Comment · Reviewer_uSST · 2025-08-07
> > > **Official Comment by Reviewer uSST**
> > >
> > > Thank you for the detailed rebuttal. While I appreciate the clarifications and additional analyses provided by authors, some of my core concerns remain:
> > >
> > > - **Fairness in Comparison:** I understand that the primary focus of Science0-c is on a narrow set of well-defined chemistry reasoning tasks. However, in that case, comparing performance against general-purpose models that were not fine-tuned on these specific tasks may not be entirely fair or informative. These models were not optimized for the target problems and may underperform due to misalignment with task formulation, rather than limitations in their reasoning ability.
> > >
> > >
> > > - **Novelty of Using Reasoning Models for Chemistry:** The claim that reasoning models are particularly suited for open-ended chemistry problems is not entirely novel. Prior work, such as [1], has already explored this direction.
> > >
> > > - **Reward Design and Evaluation Robustness:** The insights and strategies for designing reward functions could be valuable contributions. However, the manuscript does not systematically emphasize these aspects. Furthermore, since the same or similar reward functions are used for both training and evaluation, concerns remain about the robustness of the evaluation—specifically, the risk that models may overfit to reward artifacts rather than exhibit genuine reasoning competence.
> > >
> > > - **Qualitative Comparison:** I appreciate the inclusion of the qualitative trace analysis between Science0-c and DeepSeek-r1. This is a helpful addition and directly addresses my earlier concern regarding qualitative reasoning behavior.
> > >
> > >
> > > Overall, while the rebuttal addresses some issues—especially the qualitative comparison—it does not fully resolve my concerns regarding the novelty and robustness of the evaluation. I will therefore maintain my original score.
> > >
> > > [1] Mirza, Adrian, et al. “Are large language models superhuman chemists?” arXiv preprint arXiv:2404.01475 (2024)

---

> ### Author Response · Authors · 2025-08-07
>
> We appreciate the reviewer’s updated feedback, and would like to make some further clarifications
>
> **Fairness in Comparison**
>
> Our comparison strategy focuses on benchmarking against the current state-of-the-art language models for the chemistry tasks under consideration, regardless of whether they are general-purpose or domain-specific. We specifically evaluate against language models because several tasks must be formulated in natural language (MCQs, solubility editing, etc.) and non-language models cannot handle such formulations. Posing all problems in natural language further enables human-model interaction.
>
> Notably, general-purpose models like o1 and o3 currently achieve good performance on chemistry tasks (as demonstrated in their published system cards). This makes them necessary benchmarks. Not because they are general models, but because they represent the current state-of-the-art in LLMs for chemistry. Our evaluation also includes chemistry-specialized language models such as ChemDFM, Molecular Transformer (MT), and TxGemma. Note that even specialized models sometimes fall behind general-purposed models in some tasks.
>
> Therefore, we compared against the most relevant and best performant available baselines for each task in the current landscape of language models.
>
> **Novelty**
>
> We acknowledge that [1] evaluated a reasoning model (OpenAI's o1) on chemistry tasks and demonstrated strong performance compared to non-reasoning models.
>
> However, a key limitation of [1] is that o1 is a closed-source model with unknown architecture, size, and training details. When [1] compares o1 to other models (all trained with different pipelines), we cannot determine whether performance gains stem from reasoning capabilities or other factors such as a more suitable pre-training data. We are limited to the fact that o1 performs better.
>
> In contrast, our work provides controlled experimental evidence by comparing the same base model trained with and without reasoning techniques. This approach isolates the specific contribution of reasoning to performance improvements, eliminating confounding variables like model size, architecture, or training data differences. Through this controlled comparison, we definitively establish that reasoning capabilities enhance performance on open-ended scientific problems.
>
> Beyond proving this fundamental point, our work makes several additional contributions listed in our previous comment.
>
> **Reward and Evaluation**
>
> > However, the manuscript does not systematically emphasize these aspects…the risk that models may overfit to reward artifacts
>
> We acknowledge that the manuscript does not put these details in one place, but rather they are spread between Section 3, Appendix C, SI Section 2.2, and the codebase. These mitigations were designed to avoid the concern of “overfit[ting] on reward artifacts” the reviewer mentions:
>
> * Reasonable molecule check (see Section 3, C.3): we decompose the molecule into overlapping rings and fragments, and confirm that these rings and fragments pass a set of criteria derived from synthesized, well-characterized molecules. We take a very conservative approach specifically to avoid any risk of our model exploiting loopholes. We are happy to include examples of molecules that fail this check in the manuscript, to motivate its design.
>
> * Purchasability check (line 109): when proposing a synthesis pathway, the model can in principle utilize highly-contrived precursors to arrive at the derived product. We therefore require all reactants to be purchasable (i.e. purchasable). We can add this detail to the manuscript, including the specific catalogs used.
>
> * Reward functions used for IUPAC naming, reaction prediction, and MCQs are not susceptible to any artifacts/hacks.
>
> > the same or similar reward functions are used for both training and evaluation
>
> It is not uncommon to change the evaluation criterion when measuring out-of-distribution generalization, or when reward shaping is used during training. However, this work does neither (with the exception of one soft reward, which is indeed removed when evaluating).
>
> We designed a dataset of problems encountered in and derived from real chemistry and drug-design scenarios. Our model is trained to solve unseen problems from this distribution, not out-of-distribution problems.
>
> Similarly, our reward functions are designed by professional chemists to be chemically accurate and to enforce realistic constraints (see above). While we agree that independent human evaluator assessment of the designed molecules would add robustness to our result, we believe our methodology (carefully-designed metrics by human experts) is a reasonable proxy. We therefore believe it is appropriate to use the same criterion during training, evaluation, and real-world usage.
>
> **References:**
>
> [1] Mirza, Adrian, et al. “Are large language models superhuman chemists?” arXiv preprint arXiv:2404.01475 (2024)

---

### Note · Authors · 2025-08-12

We sincerely thank all reviewers for their constructive feedback. It has been very valuable to refine our paper. The discussions primarily centered on four key areas:

**Novelty and Contribution**

Reviewers uSST and ha5g inquired about the novelty of our training pipeline. While we use established techniques (SFT, GRPO), we highlight three unique contributions: (1) advantage-based curriculum training that guides learning in sparse-reward environments, (2) rephrased demonstration trajectories using the base model for improved token alignment, and (3) a multi-stage training pipeline incorporating specialist models (also incorporated by concurrent work). In addition, to the best of our knowledge, this is the first work demonstrating that reasoning helps models solve complex open-ended scientific tasks. We also provide an analysis of the model behaviour and design insights for robust chemistry reward functions that prevent reward hacking.

**Evaluation and Comparison**

Reviewers uSST and ha5g raised questions about the evaluation. We maintain that our benchmarking is thorough, covering state-of-the-art language models for chemistry, and necessarily includes both frontier general-purpose models and specialized domain-specific models. We clarify that our goal was creating a specialized chemistry model trained on specific task types, which justifies evaluating on held-out problems from the same task distribution rather than entirely novel chemistry tasks. However, we'll acknowledge in the manuscript a limited generalization to unseen chemistry tasks like ChemBench, and will add error bars to reflect performance variability of human experts.

**Qualitative Analysis and Interpretability**

Following feedback from reviewers uSST and ZRWC, we'll incorporate a human-expert comparison of reasoning traces between our model and DeepSeek-r1, showing that our model's reasoning is perceived as more genuine and faithful.

**Technical Details and Reproducibility**

Addressing feedback from reviewers hCQw, ha5g, and dvLR, we'll provide: (1) explicit discussion of Molecular Transformer limitations for retrosynthesis validation, acknowledging that specialized non-language models may still outperform on certain tasks, (2) detailed breakdown of dataset sizes and filtering processes for each training stage, (3) explicit discussion of the broader application of Science0-c in drug discovery pipelines.

We hope this clarifies our contributions and addresses the reviewers concerns.

---

### Decision · Program_Chairs · 2025-09-17

**Decision:**

Accept (poster)

**Comment:**

This paper proposes Science0-c, a 24-billion parameter LLM, designed for chemical reasoning. Science0-c is based on the Mistral-Small-24B model without additional domain pretraining but fine-tuned to understand natural language prompts and respond with chemical structures in SMILES format.

The reviewers expressed positive support for the paper and I also think that the work makes a good contribution toward reasoning for chemistry tasks. I strongly encourage the authors to incorporate all the discussion points with the reviewers in the final version.